# Adjusting Model Size in Continual Gaussian Processes: How Big is Big Enough?

**Guiomar Pescador-Barrios**
Imperial College London

**Sarah Filippi**
Imperial College London

**Mark van der Wilk**
University of Oxford

## Abstract

Many machine learning models require setting a parameter that controls their size before training, e.g. number of neurons in DNNs, or inducing points in GPs. Increasing capacity typically improves performance until all the information from the dataset is captured. After this point, computational cost keeps increasing without improved performance. This leads to the question "How big is big enough?" We investigate this problem for Gaussian processes (single-layer neural networks) in continual learning. Here, data becomes available incrementally, and the final dataset size will therefore not be known before training, preventing the use of heuristics for setting a fixed model size. We develop a method to automatically adjust model size while maintaining near-optimal performance. Our experimental procedure follows the constraint that any hyperparameters must be set without seeing dataset properties. For our method, a single hyperparameter setting works well across diverse datasets, showing that it requires less tuning compared to others.

## 1   Introduction

Continual learning aims to train models when the data arrives in a stream of batches, without storing data after it has been processed, and while obtaining predictive performance that is as high as possible at each point in time [35]. Selecting the size of the model is challenging in this setting, since typical non-continual training procedures do this by trial-and-error (cross-validation) using repeated training runs, which is not possible under our requirement of not storing any data. Selecting model size is crucial, since if the model is too small, predictive performance will suffer. One solution could be to simply make all continual learning models so large, that they will always have enough capacity, regardless of what dataset and what amount of data they will be given. However, this "worst-case" strategy is wasteful of computational resources.

A more elegant solution would be to grow the size of the model adaptively as data arrives, according to the needs of the problem (see Figure 1 for an illustration). For example, if data were only ever gathered from the same region, there would be diminishing novelty in every new batch, leading to a possible halt in growth, with growth resuming once data arrives from new regions. In this paper, we investigate a principle for determining how to select the size of a model so that it is sufficient to obtain near-optimal performance, while otherwise wasting a minimal amount of computation. In other words, we seek to answer the question of "how big is big enough?" for setting the size of models throughout continual learning.

We investigate this question for Gaussian processes where excellent continual learning methods exist but assume a fixed model capacity that is large enough. We introduce a criterion for determining the necessary number of inducing variables as new data arrives. Our method achieves near-optimal performance with fewer computational resources than other continual methods. With only one hyperparameter to balance cost and accuracy, a single value works effectively across datasets, enabling all modelling decisions to be made upfront. For related work, see App. B.

Workshop on Bayesian Decision-making and Uncertainty, 38th Conference on Neural Information Processing Systems (NeurIPS 2024).

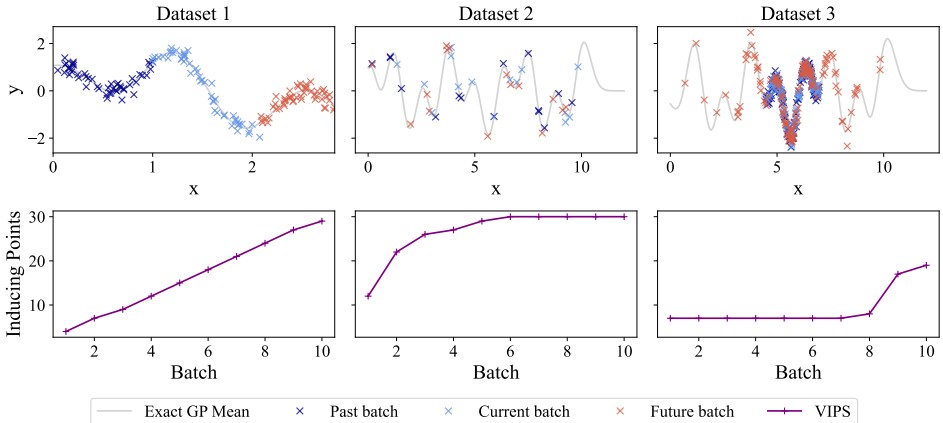

Figure 1: Three continual learning scenarios with different capacity requirements. **Top**: Three consecutive batches for 1) a growing input space 2) i.i.d. samples from a uniform distribution, and 3) narrow-range samples with occasional outliers. **Bottom**: Number of inducing points selected using the VIPS algorithm at each batch. We observe: 1) a linear increase, 2) after initial training, we see a halt in growth, and 3) low model size until it encounters outliers.

## 2 Background

### 2.1 Sparse Variational Gaussian Processes

We consider the typical regression setting, with training data consisting of $N$ input/output pairs $\{\mathbf{x}_n, y_n\}_{n=1}^N, \mathbf{x}_n \in \mathbb{R}^D, y_n \in \mathbb{R}$. We model $y_n$ by passing $\mathbf{x}_n$ through a function followed by additive Gaussian noise $y_n = f(\mathbf{x}_n) + \epsilon_n, \epsilon_n \sim \mathcal{N}(0, \sigma^2)$, and take a Gaussian process prior on $f \sim \mathcal{GP}(0, k_\theta(\cdot, \cdot))$ with zero mean, and a kernel $k$ with hyperparameters $\theta$. While the posterior (for prediction) and marginal likelihood (for finding $\theta$) can be computed in closed form [34], they have a computational cost of $O(N^3)$ that is too high, and require all training data (or statistics greater in size) to be stored, both of which are prohibitive for continual learning. Variational inference can provide an approximation at a lower $O(NM^2)$ computational and $O(NM)$ memory costs by selecting an approximation from a set of tractable posteriors

$$q(f(\cdot)) = \int p(f(\cdot)|\mathbf{u}, \theta)q(\mathbf{u})\mathrm{d}\mathbf{u} \tag{1}$$

$$= \mathcal{N}\left(f(\cdot); \mathbf{k}_{\cdot\mathbf{u}}\mathbf{K}_{\mathbf{uu}}^{-1}\mathbf{m}, k(\cdot, \cdot) - \mathbf{k}_{\cdot\mathbf{u}}\mathbf{K}_{\mathbf{uu}}^{-1}(\mathbf{K}_{\mathbf{uu}} - \mathbf{S})\mathbf{K}_{\mathbf{uu}}^{-1}\mathbf{k}_{\mathbf{u}\cdot}\right), \tag{2}$$

with $[\mathbf{K}_{\mathbf{uu}}]_{ij} = k(\mathbf{z}_i, \mathbf{z}_j), [\mathbf{k}_{\cdot\mathbf{u}}]_i = [\mathbf{k}_{\mathbf{u}\cdot}^T]_i = k(\cdot, \mathbf{z}_i), \mathbf{Z} = \{\mathbf{z}_m\}_{m=1}^M$, and $q(\mathbf{u}) = \mathcal{N}(\mathbf{u}; \mathbf{m}, \mathbf{S})$. The variational parameters $\mathbf{m}, \mathbf{S}, \mathbf{Z}$ and hyperparameters $\theta$ are selected by maximising the Evidence Lower Bound (ELBO). This simultaneously minimises KL gap $\mathrm{KL}[q(f) \| p(f|\mathbf{y}, \theta)]$ between the approximate and true GP posteriors [26, 25], and maximises an approximation to the marginal likelihood of the hyperparameters:

$$\mathcal{L}_{\mathrm{ELBO}} = \sum_{i=1}^N \mathbb{E}_{q(f(\mathbf{x}_i))}[\log p(y_i|f(\mathbf{x}_i), \theta)] - \mathrm{KL}\left[q(\mathbf{u}) \| p(\mathbf{u}|\theta)\right] . \tag{3}$$

The variational approximation has the desirable properties [44] of **1)** providing a measure of discrepancy between the finite capacity approximation, and the true infinite capacity model, **2)** resulting in arbitrarily accurate approximations if enough capacity is added [3], and **3)** retaining the uncertainty quantification over the infinite number of basis functions. In this work, we will particularly rely on being able to measure the quality of the approximation to help determine how large $M$ should be.

### 2.2 Sparse Gaussian Processes are Equivalent to Single-Layer Neural Networks

For inner product kernels $k(\mathbf{x}, \mathbf{Z}) = \sigma(\mathbf{Z}\mathbf{x})$ like the arc-cosine kernel [6], the mean is equivalent to a single-layer neural network with $\mathbf{Z}$ as the input weights, and $\mathbf{K}_{\mathbf{uu}}^{-1}\mathbf{m}$ as the output weights. This

construction also arises from other combinations of kernels and inter-domain inducing variables [9, 40], and has also shown equivalences between deep Gaussian processes and deep neural networks [10]. As a consequence, our method for determining the number of inducing variables needed in a sparse GP, equivalently finds the number of neurons needed in a single-layer neural network.

## 2.3 Online Sparse Gaussian Processes

We use the extension of the sparse variational GP approximation to the continual learning case developed by Bui et al. [2]. We update our posterior and hyperparameter approximations after each batch of new data $\{\mathbf{X}_n, \mathbf{y}_n\}$. While we do not have access to data from older batches $\{\mathbf{X}_o, \mathbf{y}_o\}$, the parameters specifying the approximate posterior $q_o(f) = p(f_{\neq \mathbf{a}}|\mathbf{a}, \theta_o) q_o(\mathbf{a})$ are passed on. This approximate posterior is constructed as in eq. (1) but with $\mathbf{a} = f(\mathbf{Z}_o)$ and the old hyperparameters $\theta_o$. Given the "old" $q_o(f)$, online sparse GPs construct a "new" approximation $q_n(f) = p(f_{\neq \mathbf{b}}|\mathbf{b}, \theta_n) q_n(\mathbf{b})$, where $\mathbf{b} = f(\mathbf{Z}_n)$ and $\theta_n$ is the new hyperparameter, of the posterior distribution for all observed data $p(f|\mathbf{y}_o, \mathbf{y}_n, \theta_n)$. This is done by maximising the following the training objective:

$$\widehat{\mathcal{L}} := \int q_n(f) \left[ \log \frac{p(\mathbf{b}|\theta_n) q_o(\mathbf{a}) p(\mathbf{y}_n|f)}{q_n(\mathbf{b}) p(\mathbf{a}|\theta_o)} \right] \mathrm{d}f, \tag{4}$$

which we refer to as the "online ELBO". We provide technical details of this quantity in App. B.1, where we modify the typical derivation to **1)** clarify how the online ELBO provides an estimate to the full-batch ELBO, and **2)** clarify when this approximation is accurate.

To achieve a fully black-box solution, we must specify how to choose the hyperparameters $\theta_n$, the number of inducing variables $M_\mathbf{b}$, and the inducing inputs $\mathbf{Z}_n$. We select $\theta_n$ by maximising $\widehat{\mathcal{L}}$ using L-BFGS and determine $\mathbf{Z}_n$ using the "greedy variance" criterion [12, 13, 3]. This leaves only the number of inducing variables $M_\mathbf{b}$ to be chosen.

# 3 Automatically Adapting Approximation Capacity

We propose a method for adjusting the capacity of the approximation $M_\mathbf{b}$ to maintain accuracy. We keep inducing points from old batches fixed, and select new inducing points from each incoming batch, with their locations set using the "greedy variance" criterion [3, 12, 13]. While optimising all inducing points leads to a strictly better approximation, we avoid this for simplicity. The question remains: To achieve a certain level of accuracy, "how big is big enough?" To answer this, we consider the online ELBO as a function of the capacity $\widehat{\mathcal{L}}(M_\mathbf{b})$, and propose a threshold after which to stop adding new inducing variables.

## 3.1 Online Log Marginal Likelihood (LML) Upper Bound

The problem of selecting enough inducing variables remains open in the full-batch setting. One possible strategy is to derive an upper bound on the marginal likelihood ($\mathcal{U}$) and stop adding inducing variables the difference $\mathcal{U} - \mathcal{L}$ (which upper bounds $\mathrm{KL}[q(f)||p(f|\mathbf{y})]$) falls below a tolerance $\alpha$ [42]. Similarly, we consider the maximum possible value of our lower bound, which in the online setting is obtained by retaining previous inducing inputs and adding each new datapoint to the inducing set:

$$\mathcal{L}^* := \widehat{\mathcal{L}}(N_n + M_\mathbf{a}) = \log \mathcal{N}\left(\hat{\mathbf{y}}; \mathbf{0}, \mathbf{K}_{\hat{\mathbf{f}}\hat{\mathbf{f}}} + \Sigma_{\hat{y}}\right) + \Delta_\mathbf{a} \quad \text{with} \quad \mathbf{K}_{\hat{\mathbf{f}}\hat{\mathbf{f}}} = \begin{bmatrix} \mathbf{K}_{\mathbf{ff}} & \mathbf{K}_{\mathbf{fa}} \\ \mathbf{K}_{\mathbf{af}} & \mathbf{K}_{\mathbf{aa}} \end{bmatrix}. \tag{5}$$

Using properties of positive semi-definite matrices, we derive an upper bound $\widehat{\mathcal{U}}(M)$ to eq. (5):

$$\mathcal{L}^* \leq -\frac{(N_n + M_\mathbf{a})}{2} \log(2\pi) - \frac{1}{2} \log |\mathbf{Q}_{\hat{\mathbf{f}}\hat{\mathbf{f}}} + \Sigma_{\hat{\mathbf{y}}}| - \frac{1}{2} \hat{\mathbf{y}}^T \left( \mathbf{Q}_{\hat{\mathbf{f}}\hat{\mathbf{f}}} + t\mathrm{I} + \Sigma_{\hat{\mathbf{y}}} \right)^{-1} \hat{\mathbf{y}} + \Delta_\mathbf{a} := \widehat{\mathcal{U}}(M),$$

where $t = \mathrm{tr}(\mathbf{K}_{\hat{\mathbf{f}}\hat{\mathbf{f}}} - \mathbf{Q}_{\hat{\mathbf{f}}\hat{\mathbf{f}}})$ and $\mathbf{Q}_{\hat{\mathbf{f}}\hat{\mathbf{f}}} = \mathbf{K}_{\hat{\mathbf{f}}\mathbf{b}} \mathbf{K}_{\mathbf{bb}}^{-1} \mathbf{K}_{\mathbf{b}\hat{\mathbf{f}}}$ and $M$ is the number of inducing points used to calculate the bound (which can be unequal to $M_\mathbf{b}$).

## 3.2 Approximation Quality Guarantees

Adding inducing points will eventually increase $\widehat{\mathcal{L}}$ until it reaches $\mathcal{L}^*$ [1, 25, 3]. If we add inducing points until $\widehat{\mathcal{U}}(M) - \widehat{\mathcal{L}}(M_\mathbf{b}) \leq \alpha$ we can guarantee the following:

**Guarantee.** *Let $M$ be a fixed integer and $M_{\mathbf{b}}$ be the number of selected inducing points such that $\widehat{\mathcal{U}}(M) - \widehat{\mathcal{L}}(M_{\mathbf{b}}) \leq \alpha$. Assuming that $\theta_n = \theta_o$, we have two equivalent bounds:*

$$\mathrm{KL}[q_n(f) \,||\, p(f|\mathbf{y}_o, \mathbf{y}_n, \theta_o)] \leq \alpha + \Psi \tag{6}$$

$$\mathrm{KL}[q_n(f) \,||\, q_n^*(f)] \leq \alpha \tag{7}$$

*where $\Psi = \int q_n(f) \log \frac{q_n^*(f)}{p(f|\mathbf{y}_n, \mathbf{y}_o)} \mathrm{d}f$ and $q_n^*(f) = \mathcal{Z}^{-1} q_o(f) p(\mathbf{y}_n \mid f)$ represents the variational distribution associated with the optimal lower bound $\mathcal{L}^* = \widehat{\mathcal{L}}(N_n + M_{\mathbf{a}})$, with $\mathcal{Z}$ denoting the marginal likelihood that normalises $q_n^*(f)$.*

*Proof.* We cease the addition of points when $\widehat{\mathcal{L}}(M_{\mathbf{b}}) \geq \widehat{\mathcal{U}}(M) - \alpha$. Given that $\widehat{\mathcal{U}}(M) \geq \mathcal{L}^*$, and assuming $\theta_n = \theta_o$, the rest follows from algebraic manipulation of eq. (9). See App. C for the complete proof. $\qquad\square$

The first bound shows that if $\Psi$ is near zero, the KL to the true posterior is bounded by $\alpha$. While $\Psi$ depends on the true posterior and therefore cannot be computed, if the posterior in the previous iteration was exact, $\Psi$ would be equal to zero. The second bound shows that we are guaranteed to have our actual approximation $q_n(f)$ be within $\alpha$ nats of the best approximation that we can develop, given the limitations of the approximations made in previous iterations.

### 3.3 Selecting a Threshold

In this final step of our online learning method, we must specify a heuristic for selecting $\alpha$ that does not require knowing any data in advance, while also working in a uniform way across datasets with different properties. A constant value for $\alpha$ does not work well, since the scale of the LML depends strongly on properties such as dataset size, and observation noise. This means that a tolerance of 1 nat [7] may be appropriate for a small dataset, but not for a large one.

As a principle for selecting the threshold, we take loose inspiration from compression and MDL [18], which takes the view of the ELBO being proportional to negative the code length that the model requires to encode the dataset. Intuitively, our desire is to select an $\alpha$ such that our method captures a high proportion (e.g. 95%) of all the information in each batch, so that we can compress to within a small fraction of the optimal variational code. To address the issue of undefined quantisation tolerance, we use an independent random noise code as our baseline and choose $\alpha$ to be within a small fraction of the optimal variational code relative to the random noise code. We want to be able to capture a high proportion of the additional information provided by our model relative to the noise model, i.e. we want our threshold to be:

$$\alpha = \delta(\mathcal{L}^* - \mathcal{L}_{\mathrm{noise}}), \quad \mathcal{L}_{\mathrm{noise}} = \sum_{n=1}^{N_n} \log \mathcal{N}(y_n; \hat{\mu}, \hat{\sigma}^2)$$

where $\hat{\mu}$ and $\hat{\sigma}^2$ are the average and variance of the observations for up to the current task and $\delta$ is a user-defined hyperparameter. We validate that this approach leads to values of $\delta$ giving consistent behaviour across a wide range of datasets, which allows it to be set in advance without needing much prior knowledge of the dataset characteristics.

Calculating this threshold is intractable for large batch sizes $N_n$. However, if we change our stopping criterion to the more stringent upper bound

$$\bar{\alpha} = \delta(\widehat{\mathcal{U}}(M) - \mathcal{L}_{\mathrm{noise}}) \tag{8}$$

and increase $M$ for calculating $\widehat{\mathcal{U}}$ as $M_{\mathbf{b}}$ is increased for calculating $\widehat{\mathcal{L}}$, we obtain the same guarantees as before but at a lower computational cost. However, this strategy is only worthwhile for very large batch sizes $N_n$, due to the importance of constant factors in the computational cost. In the common continual learning settings we investigate $N_n$ is small enough to allow computing $\mathcal{L}^*$.

The algorithm for our inducing point selection method can be found in App. D. We name our approach Vegas Inducing Point Selection (VIPS), drawing an analogy to Las Vegas Algorithms. These methods guarantee the accuracy of the output, however, their computational time fluctuates for every run [29].

Table 1: Mean (std) over different training/test splits of the number of inducing points for the last batch for operating point (sec 4.2). The cross (✗) denotes unmet accuracy constraint, while "Max." indicates that maximum capacity was reached.

| UCI Dataset | Dimension (N, D) | Conditional Variance (CV) | OIPS [14] | VIPS (Ours) |
|---|---|---|---|---|
| Concrete | 1030, 8 | 461(59) | 409(87) | **385(84)** |
| Skillcraft | 3338, 19 | 599(30) | 332(82) | **141(4)** |
| Kin8nm | 8192, 8 | 6194(13) | 6539(9) | **2953(72)** |
| Naval | 11934, 14 | **35(3)** | ✗ | 127(5) |
| Elevators | 16599, 18 | 2501(100) | 643(135) | **332(8)** |
| Bike | 17379, 17 | Max. 7000 | 5131(65) | **1037(24)** |

# 4 Experiments

We evaluate the performance of our adaptive inducing point selection, VIPS, in a range of streaming scenarios where we assume the total number of observations is unknown. In all cases, the variational distribution and kernel hyperparameters are optimised using the online lower bound (Eq. (13)).

Continual learning scenarios pose unique challenges: memory allocation cannot be pre-determined due to unknown input space coverage, and cross-validation for hyperparameter tuning is infeasible as it requires storing all data. Thus, an effective method must **1)** have an adaptive memory that can grow with the demands of the data, **2)** work with hyperparameters that can be set before training. Our experiments aim to illustrate these points. Details and additional experiments are provided in App. F.

## 4.1 Model size and data distribution

Figure 1 shows VIPS's ability to adapt across datasets with different characteristics, each divided into ten batches, illustrating how input distribution drives model growth as more data is seen. In the first dataset, each batch introduces new data, causing the model size to grow linearly. The second dataset remains within a fixed interval, leading to reduced novelty in batches and a converging model size. The third dataset combines narrow-range samples with occasional outliers, resulting in low model size with occasional growth when novelty appears (details in App. F.1).

## 4.2 Continual learning of UCI datasets

We compare VIPS to two other inducing point selection methods: Conditional Variance (CV) and OIPS [14] (details in App. E). We use six datasets from the UCI repository [8], simulating a continual learning scenario by sorting the data along the first dimension and dividing it into batches. For each method, we assess multiple hyperparameter settings and identify the one that minimises model size while achieving RMSE within 10% of a full-batch GP across all datasets, considered equivalent to near-exact performance. Table 1 shows the number of inducing points used for that particular hyperparameter value. The method CV often leads to larger model sizes (excessive for "bike"). For the noiseless "naval" dataset, CV uses fewer inducing points but obtains poor uncertainty estimates (details in App. F.3.2) and OIPS fails to meet the accuracy constraint within its tested hyperparameter range. Meanwhile, VIPS consistently meets accuracy requirements and uses fewer inducing points in the majority of datasets, suggesting it requires less hyperparameter tuning (details in App. F.3).

# 5 Discussion

In this work, we propose a method to dynamically adjust the number of inducing variables in streaming GP regression, providing a capacity control criterion with approximation guarantees. Our method achieves a performance close to full-batch approaches while minimising model size. It relies on a single hyperparameter to balance accuracy and complexity, and we demonstrate that a single setting performs well across diverse datasets. This reduces the need for extensive hyperparameter tuning and eliminates the requirement to pre-define model size, thereby addressing a significant bottleneck in traditional methods. While our current focus is on GPs, we aim to extend this method to larger neural architectures.

## Acknowledgments and Disclosure of Funding

GPB is supported by EPSRC through the Statistical Machine Learning (StatML) CDT programme, grant no. EP/S023151/1.

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

# A   Code

The methods discussed in this work, along with the code to reproduce our results, are available online at https://github.com/guiomarpescador/vips.

# B   Related Work

The most widely discussed problem in continual learning is that of *catastrophic forgetting*, where previously acquired knowledge is lost in favour of recent information [28, 17]. Many solutions have been proposed in the literature, such as encouraging weights to be close to values that were well-determined by past data [21, 38], storing subsets or statistics of past data to continue to train the neural network in the future [22, 23], and approximate Bayesian methods that balance uncertainty estimates of parameters with the strength of the data [32, 36, 5]. Within continual learning, many different settings have been investigated, which vary in difficulty [11]. Across these tasks, the gap in performance to a full-batch training procedure therefore also varies, but despite progress, some gap in performance remains.

Bayesian continual learning methods have been developed because the posterior given past data becomes the prior for future data, making the posterior a sufficient quantity to estimate [30]. For the special case of linear-in-the-parameters regression models, the posterior and updates can be calculated in closed form, leading to continual learning giving *exactly* the same result as full-batch training. In most cases (e.g. for neural networks), the posterior cannot be found exactly, leading to the aforementioned methods [32, 36, 5] that focus on finding an approximation to the posterior and using this as the sufficient quantity.

Even with a perfect solution to catastrophic forgetting (e.g. in the case of linear-in-the-parameters regression models), continual learning methods face the additional difficulty of ensuring that models have sufficient capacity to accommodate the continuously arriving information. In continual learning, it is particularly difficult to determine a fixed size for the model, since the number of data or tasks are not yet known, and selecting a model that is too small can significantly hurt performance. To improve over a fixed model size, methods can be made to *grow* with the data size. For example, Rusu et al. [37] extend hidden representations by a fixed amount for each new batch of data that arrives, and allows the weights of the extended representation to depend on the representation of all previous tasks. Yoon et al. [45] argue that extension by a fixed amount is wasteful and should instead be data dependent, specifically by copying neurons if their value changes too much, and adding new neurons if the training loss doesn't reach a particular threshold. Kessler et al. [20] propose to use the Indian Buffet Process as a more principled way to regularise how fast new weights are added with tasks. While the data dependence that both these methods introduce is necessary to prevent computational waste, both methods have hyperparameters that need to be tuned to dataset characteristics, which is difficult when the dataset characteristics are not known at the start of training.

Growing model capacity with dataset size was one of the main justifications for research into (Bayesian) non-parametric models [15, 16]. This approach defines models with infinite capacity, with Bayesian inference naturally using an appropriate finite capacity to make predictions, with finite compute budgets. Gaussian processes (GPs) [34] are the most common Bayesian non-parametric model for supervised learning, and are equivalent to infinitely-wide deep neural networks [31, 27] and linear-in-the-parameters models with an infinite feature space. Their infinite capacity allows them to recover functions perfectly in the limit of infinite data [43], and their posterior can be computed in closed form. These two mathematical properties provide strong principles for providing high-quality solutions to both catastrophic forgetting and ensuring appropriate capacity, and therefore make GPs an excellent model for studying continual learning.

However, developing practical continual learning in GPs is not as straightforward as it is in finite dimensional linear models, because (for $N$ datapoints) the posterior requires **1)** $O(N^3)$ operations to compute it exactly, which becomes intractable for large datasets, and **2)** storing the full training dataset, which breaks the requirements of continual learning. Sparse variational inducing variable methods have been proposed to solve these problems [41], by introducing a small number of $M$ inducing points that control the capacity of the posterior approximation. In certain settings, this approximation is near-exact even when $M \ll N$ [3]. This property has allowed continual learning methods to be developed for GPs that perform very closely to full-batch methods [2, 24, 4], *provided* $M$ is large enough.

As in neural network models, selecting the capacity $M$ is an open problem, with several proposed solutions. Kapoor et al. [19] acknowledge the need for scaling the capacity with data size, and propose VAR-GP (Variational Autoregressive GP) which adds a fixed number of inducing points for every batch. However, this number may be too small, leading to poor performance, or too large, leading to wasted computation. Galy-Fajou and Opper [14] propose OIPS (online inducing point selection), which determines $M$ through a threshold on the correlation with other inducing points, which needs to be tuned based on dataset properties.

In this work, we propose to instead select the capacity of the variational approximation by selecting an appropriate tolerance in the KL gap to the true posterior. This criterion works within the same computational constraints as existing GP continual learning methods, adapts the capacity to the dataset to minimise computational waste while retaining near-optimal performance. Our method has a single hyperparameter that we keep fixed to a single value, and that produces similar trade-offs across benchmark datasets with significantly different characteristics.

## B.1 Online Sparse Gaussian Processes

In this work, we use the extension of the sparse variational GP approximation to the continual learning case developed by Bui et al. [2]. We modified the typical derivation to **1)** clarify how the online ELBO provides an estimate to the full-batch ELBO, and **2)** clarify when this approximation is accurate.

In this online setting, we aim to update our posterior and hyperparameter approximations after each batch of new data $\{\mathbf{X}_n, \mathbf{y}_n\}$. While we do not have access to data from older batches $\{\mathbf{X}_o, \mathbf{y}_o\}$, the parameters specifying the approximate posterior $q_o(f) = p(f_{\neq \mathbf{a}}|\mathbf{a}, \theta_o)q_o(\mathbf{a})$ are passed on. This approximate posterior is constructed as in eq. (1) but with $\mathbf{a} = f(\mathbf{Z}_o)$ and the old hyperparameters $\theta_o$. Given the "old" $q_o(f)$, online sparse GPs construct a "new" approximation $q_n(f)$ of the posterior for all observed data $p(f|\mathbf{y}_o, \mathbf{y}_n, \theta_n)$, which can be written as:

$$p(f|\mathbf{y}_o, \mathbf{y}_n, \theta_n) = \frac{p(f|\theta_n)p(\mathbf{y}_n|f)p(\mathbf{y}_o|f)}{p(\mathbf{y}_n, \mathbf{y}_o|\theta_n)} = \frac{p(f|\theta_n)p(\mathbf{y}_n|f)}{p(\mathbf{y}_n, \mathbf{y}_o|\theta_n)} \frac{p(f|\mathbf{y}_o, \theta_o)p(\mathbf{y}_o|\theta_o)}{p(f|\theta_o)} .$$

We denote the new variational distribution as $q_n(f) = p(f_{\neq \mathbf{b}}|\mathbf{b}, \theta_n)q_n(\mathbf{b})$ where $\mathbf{b} = f(\mathbf{Z}_n)$ and $\theta_n$ is the new hyperparameter which can differ from $\theta_o$. The KL divergence between the exact and approximate posterior at the current batch is given by:

$$\text{KL}[q_n(f) \,||\, p(f|\mathbf{y}_o, \mathbf{y}_n, \theta_n)] = \log \frac{p(\mathbf{y}_n, \mathbf{y}_o|\theta_n)}{p(\mathbf{y}_o|\theta_o)} - \int q_n(f) \log \frac{p(f|\theta_n)p(\mathbf{y}_n|f)p(f|\mathbf{y}_o, \theta_o)}{q_n(f)p(f|\theta_o)} \mathrm{d}f .$$

The posterior distribution $p(f|\mathbf{y}_o, \theta_o)$ is not available, however by multiplying its approximation $q_o(f)$ in both sides of the fraction inside the log, we obtain:

$$\text{KL}[q_n(f) \,||\, p(f|\mathbf{y}_o, \mathbf{y}_n, \theta_n)] = \log \frac{p(\mathbf{y}_n, \mathbf{y}_o|\theta_n)}{p(\mathbf{y}_o|\theta_o)} - \int q_n(f) \log \frac{p(f|\theta_n)p(\mathbf{y}_n|f)q_o(f)}{q_n(f)p(f|\theta_o)} \mathrm{d}f + \Phi \quad (9)$$

where $\Phi = - \int q_n(f) \log \frac{p(f|\mathbf{y}_o, \theta_o)}{q_o(f)} \mathrm{d}f$. We cannot compute $\Phi$ due to its dependence on the exact posterior, so we drop it and use the following "online ELBO" as our training objective:

$$\widehat{\mathcal{L}} := \int q_n(f) \left[ \log \frac{p(\mathbf{b}|\theta_n)q_o(\mathbf{a})p(\mathbf{y}_n|f)}{q_n(\mathbf{b})p(\mathbf{a}|\theta_o)} \right] \mathrm{d}f. \quad (10)$$

Maximising $\widehat{\mathcal{L}}$ will accurately minimise the KL to the true posterior when $\Phi$ is small, which is the case when the old approximation is accurate, i.e. $q_o(f) \approx p(f|\mathbf{y}_o, \theta_o)$ for all values of $f$ (with

$\Phi = 0$ in the case of equality). In our continual learning procedure, we will keep our sequence of approximations accurate by ensuring they all have enough inducing points.

To get our final bound, we perform a change of variables for the variational distribution $q_o(\mathbf{a}) = \mathcal{N}(\mathbf{a}; \mathbf{m_a}, \mathbf{S_a})$ to use the likelihood parametrisation [33]:

$$q_o(\mathbf{a}) = \frac{\mathcal{N}(\mathbf{a}; \tilde{\mathbf{m}}_\mathbf{a}, \mathbf{D_a})\mathcal{N}(\mathbf{a}; 0, \mathbf{K}'_{\mathbf{aa}})}{\int \mathcal{N}(\mathbf{a}; \tilde{\mathbf{m}}_\mathbf{a}, \mathbf{D_a})\mathcal{N}(\mathbf{a}; 0, \mathbf{K}'_{\mathbf{aa}})d\mathbf{a}} = \frac{l(\mathbf{a})p(\mathbf{a} \mid \theta_o)}{\mathcal{N}(\mathbf{a}; 0, \mathbf{D_a} + \mathbf{K}'_{\mathbf{aa}})}, \tag{11}$$

where $\mathbf{D_a} = \left(\mathbf{S_a}^{-1} - \mathbf{K}'^{-1}_{\mathbf{aa}}\right)^{-1}$ and $\tilde{\mathbf{m}}_\mathbf{a} = \mathbf{K}'^{-1}_{\mathbf{aa}}\mathbf{m_a}$ are the variational parameters, $\mathbf{K}'_{\mathbf{aa}}$ is the covariance for the prior distribution $p(\mathbf{a} \mid \theta_o)$ and $l(\mathbf{a}) := \mathcal{N}(\mathbf{a}; \tilde{\mathbf{m}}_\mathbf{a}, \mathbf{D_a})$. In this formulation, the variational parameters $\tilde{\mathbf{m}}_\mathbf{a}, \mathbf{D_a}$ effectively form a dataset that produce the same posterior as the original dataset, but which we have chosen to be smaller in size, $M < N$. This makes our online ELBO from eq. (10)

$$\widehat{\mathcal{L}} = \mathbb{E}_{q_n(f)}\left[\log p(\mathbf{y}_n|f)\right] + \mathbb{E}_{q_n(f)}\left[\log l(\mathbf{a})\right] - \mathrm{KL}\left[q_n(\mathbf{b}) \;||\; p(\mathbf{b}|\theta_n)\right] - \log \mathcal{N}(\mathbf{a}; 0, \mathbf{K}'_{\mathbf{aa}} + \mathbf{D_a}), \tag{12}$$

which has the nice interpretation of being the normal ELBO, but with an additional term that includes the approximate likelihood $l(\mathbf{a})$ which summarises the effect of all previous data.

While $\widehat{\mathcal{L}}$ is all that is needed to train the online approximation, it differs from the true marginal likelihood by the term $\log p(\mathbf{y}_o|\theta_o)$. To approximate it, we could drop the term $\log \mathcal{N}(\mathbf{a}; 0, \mathbf{K}'_{\mathbf{aa}} + \mathbf{D_a})$ from $\widehat{\mathcal{L}}$, since this term also approximates $\log p(\mathbf{y}_o|\theta_o)$, with equality when the posterior is exact, but with no guarantee of being a lower bound.

Although $\widehat{\mathcal{L}}$ is a useful training objective for general likelihoods, the regression case we consider allows us to analytically find $q(\mathbf{b})$ (refer to Bui et al. [2] for derivations) resulting in the lower bound

$$\widehat{\mathcal{L}} = \log \mathcal{N}\left(\hat{\mathbf{y}}; \mathbf{0}, \mathbf{K}_{\hat{\mathbf{f}}\mathbf{b}}\mathbf{K}^{-1}_{\mathbf{bb}}\mathbf{K}_{\mathbf{b}\hat{\mathbf{f}}} + \Sigma_{\hat{\mathbf{y}}}\right) + \Delta_\mathbf{a} - \frac{1}{2}\mathrm{tr}\left[\mathbf{D}^{-1}_\mathbf{a}(\mathbf{K}_{\mathbf{aa}} - \mathbf{Q}_{\mathbf{aa}})\right] - \frac{1}{2\sigma^2}\mathrm{tr}(\mathbf{K}_{\mathbf{ff}} - \mathbf{Q}_{\mathbf{ff}}), \tag{13}$$

$$\hat{\mathbf{y}} = \left[\begin{array}{c} \mathbf{y}_n \\ \mathbf{D_a}\mathbf{S}^{-1}_\mathbf{a}\mathbf{m_a} \end{array}\right], \mathbf{K}_{\hat{\mathbf{f}}\mathbf{b}} = \left[\begin{array}{c} \mathbf{K}_{\mathbf{fb}} \\ \mathbf{K}_{\mathbf{ab}} \end{array}\right], \Sigma_{\hat{\mathbf{y}}} = \left[\begin{array}{cc} \sigma^2_{\hat{y}}\mathrm{I} & \mathbf{0} \\ \mathbf{0} & \mathbf{D_a} \end{array}\right], \tag{14}$$

$$\Delta_\mathbf{a} = -\frac{1}{2}\log \frac{|\mathbf{S_a}|}{|\mathbf{K}'_{\mathbf{aa}}||\mathbf{D_a}|} + \frac{M_\mathbf{a}}{2}\log(2\pi) - \frac{1}{2}\mathbf{m}^T_\mathbf{a}\mathbf{S}^{-1}_\mathbf{a}\mathbf{m_a} + \frac{1}{2}\mathbf{m}^T_\mathbf{a}\mathbf{S}^{-1}_\mathbf{a}\mathbf{D_a}\mathbf{S}^{-1}_\mathbf{a}\mathbf{m_a}, \tag{15}$$

with $\mathbf{Q}_{\mathbf{ff}} = \mathbf{K}_{\mathbf{fb}}\mathbf{K}^{-1}_{\mathbf{bb}}\mathbf{K}_{\mathbf{bf}}$ and $\mathbf{Q}_{\mathbf{aa}} = \mathbf{K}_{\mathbf{ab}}\mathbf{K}^{-1}_{\mathbf{bb}}\mathbf{K}_{\mathbf{ba}}$. All covariances are computed using the new hyperparameters $\theta_n$, except for $\mathbf{K}'_{\mathbf{aa}}$ which is the covariance for the prior distribution $p(\mathbf{a} \mid \theta_o)$. Finally, $M_\mathbf{a} = |\mathbf{a}|$ is the number of inducing points used at the previous batch. The computational complexity and memory requirements for calculating $\widehat{\mathcal{L}}$ at each batch is $O(N_n M^2_\mathbf{b} + M^3_\mathbf{b})$ and $O(M^2_\mathbf{b})$ respectively where $M_\mathbf{b}$ is the total number of inducing points for the current batch.

## C  Proof of Guarantee

**Guarantee.** *Let $M$ be a fixed integer and $M_\mathbf{b}$ be the number of selected inducing points such that $\widehat{\mathcal{U}}(M) - \widehat{\mathcal{L}}(M_\mathbf{b}) \le \alpha$. Assuming that $\theta_n = \theta_o$, we have two equivalent bounds:*

$$\mathrm{KL}[q_n(f) \;||\; p(f|\mathbf{y}_o, \mathbf{y}_n, \theta_o)] \le \alpha + \Psi \tag{16}$$

$$\mathrm{KL}[q_n(f) \;||\; q^*_n(f)] \le \alpha \tag{17}$$

*where $\Psi = \int q_n(f)\log \frac{q^*_n(f)}{p(f|\mathbf{y}_n, \mathbf{y}_o)}df$ and $q^*_n(f) = \mathcal{Z}^{-1}q_o(f)p(\mathbf{y}_n|f)$ represents the variational distribution associated with the optimal lower bound $\mathcal{L}^* = \widehat{\mathcal{L}}(N_n + M_\mathbf{a})$, with $\mathcal{Z}$ denoting the marginal likelihood that normalises $q^*_n(f)$.*

*Proof.* We cease the addition of points when $\widehat{\mathcal{U}}(M) - \widehat{\mathcal{L}}(M_\mathbf{b}) < \alpha$. Since $\widehat{\mathcal{U}}(M) \ge \mathcal{L}^*$, then $-\widehat{\mathcal{L}}(M_\mathbf{b}) < \alpha - \widehat{\mathcal{U}}(M) < \alpha - \mathcal{L}^*$. Eq.(4) can be bounded as:

$$\mathrm{KL}[q_n(f) \;||\; p(f|\mathbf{y}_o, \mathbf{y}_n, \theta_n)] = \log \frac{p(\mathbf{y}_n, \mathbf{y}_o|\theta_n)}{p(\mathbf{y}_o|\theta_o)} - \widehat{\mathcal{L}} + \Phi$$

$$\le \log \frac{p(\mathbf{y}_n, \mathbf{y}_o|\theta_n)}{p(\mathbf{y}_o|\theta_o)} + \alpha - \mathcal{L}^* + \Phi \tag{18}$$

where $\Phi = -\int q_n(f) \log \frac{p(f|\mathbf{y}_o, \theta_o)}{q_o(f)} \mathrm{d}f$. Let $q_n^*(f) = \mathcal{Z}^{-1} q_o(f) p(\mathbf{y}_n|f)$ be the variational distribution associated with $\mathcal{L}^* = \widehat{\mathcal{L}}(N_n + M_\mathbf{a})$. Then, by expanding the true posterior and multiplying by the variational distributions $q_n^*(f)$ on both sides of the fraction inside the log, we obtain:

$$
\begin{aligned}
\Phi &= \int q_n(f) \log \frac{q_o(f)}{p(f|\mathbf{y}_o, \theta_o)} \mathrm{d}f \\
&= \int q_n(f) \log \frac{q_o(f) p(\mathbf{y}_o|\theta_o)}{p(\mathbf{y}_o|f) p(f|\theta_o)} \mathrm{d}f \\
&= \int q_n(f) \log \frac{q_o(f) p(\mathbf{y}_o|\theta_o)}{p(\mathbf{y}_o|f) p(f|\theta_o)} \frac{q_n^*(f)}{q_n^*(f)} \mathrm{d}f \\
&= \int q_n(f) \log \frac{\cancel{q_o(f)} p(\mathbf{y}_o|\theta_o)}{p(\mathbf{y}_o|f) p(f|\theta_o)} \frac{q_n^*(f)}{\mathcal{Z}^{-1} \cancel{q_o(f)} p(\mathbf{y}_n|f)} \mathrm{d}f \\
&= \int q_n(f) \log \frac{p(\mathbf{y}_o|\theta_o) q_n^*(f)}{p(f|\mathbf{y}_n, \mathbf{y}_o, \theta_o) p(\mathbf{y}_n, \mathbf{y}_o|\theta_o)} \mathrm{d}f + \log \mathcal{Z}. \\
&= \int q_n(f) \log \frac{q_n^*(f)}{p(f|\mathbf{y}_n, \mathbf{y}_o, \theta_o)} \mathrm{d}f + \log \mathcal{Z} - \log \frac{p(\mathbf{y}_n, \mathbf{y}_o|\theta_o)}{p(\mathbf{y}_o|\theta_o)}
\end{aligned}
\tag{19}
$$

Using the above expansion for $\Phi$, eq. (18) becomes,

$$
\begin{aligned}
&\mathrm{KL}[q_n(f) \,||\, p(f|\mathbf{y}_o, \mathbf{y}_n, \theta_n)] \\
&\leq \log \frac{p(\mathbf{y}_n, \mathbf{y}_o|\theta_n)}{p(\mathbf{y}_o|\theta_o)} + \alpha - \mathcal{L}^* + \Phi \\
&\leq \log \frac{p(\mathbf{y}_n, \mathbf{y}_o|\theta_n)}{p(\mathbf{y}_o|\theta_o)} + \alpha - \mathcal{L}^* + \int q_n(f) \log \frac{q_n^*(f)}{p(f|\mathbf{y}_n, \mathbf{y}_o, \theta_o)} \mathrm{d}f + \log \mathcal{Z} - \log \frac{p(\mathbf{y}_n, \mathbf{y}_o|\theta_o)}{p(\mathbf{y}_o|\theta_o)}.
\end{aligned}
\tag{20}
$$

Assuming that $\theta_n = \theta_o$, the above can be simplified to

$$
\mathrm{KL}[q_n(f) \,||\, p(f|\mathbf{y}_o, \mathbf{y}_n, \theta_n)] \leq \alpha + \int q_n(f) \log \frac{q_n^*(f)}{p(f|\mathbf{y}_n, \mathbf{y}_o, \theta_n)} \mathrm{d}f
\tag{21}
$$

Again by multiplying by $q_n(f)$ both sides of the fraction inside the log, we obtain:

$$
\begin{aligned}
\mathrm{KL}[q_n(f) \,||\, p(f|\mathbf{y}_o, \mathbf{y}_n, \theta_o)] &\leq \int q_n(f) \log \frac{q_n^*(f)}{p(f|\mathbf{y}_n, \mathbf{y}_o, \theta_n)} \frac{q_n(f)}{q_n(f)} \mathrm{d}f + \alpha \\
\mathrm{KL}[q_n(f) \,||\, p(f|\mathbf{y}_o, \mathbf{y}_n, \theta_o)] &\leq \mathrm{KL}[q_n(f) \,||\, p(f|\mathbf{y}_o, \mathbf{y}_n, \theta_n)] - \mathrm{KL}[q_n(f) \,||\, q_n^*(f)] + \alpha \\
\mathrm{KL}[q_n(f) \,||\, q_n^*(f)] &\leq \alpha.
\end{aligned}
\tag{22}
$$

$\square$

## D  Vegas Inducing Point Selection (VIPS) Algorithm

To select the location of our new inducing points we use the location selection strategy "greedy variance" proposed in [3]. This strategy iteratively selects points from a set based on a preference criterion until a stopping condition is met. In particular, it chooses the location of the next inducing point to maximise the marginal variance in the conditional prior $p(f_{\neq \mathbf{u}}|\mathbf{u})$. This is equivalent to maximising $\mathrm{diag}[\mathbf{K_{ff}} - \mathbf{Q_{ff}}]$. In continual learning, Chang et al. [4] use the "greedy variance" criterion by defining $\{\mathbf{Z}_o, \mathbf{X}_n\}$ as the selection pool from which inducing point locations are selected and maintaining a fixed number of inducing points. Similarly, Maddox et al. [24] extends the "greedy variance" criterion to heteroskedastic Gaussian likelihoods and also uses a fixed size approach. In our case, we tested the location strategy with our stopping criterion using both $\{\mathbf{Z}_o, \mathbf{X}_n\}$ and $\{\mathbf{X}_n\}$ as candidates pool for the locations of the inducing points. We did not find a substantial difference between the methods and hence opted for the simpler version where we keep the old inducing point locations fixed and choose the new set of inducing points from among the locations in $\mathbf{X}_n$.

Algorithm 1 presents an inducing point selection method using our stopping criterion combined with the location selection strategy "greedy variance". The method takes as input a value for the

hyperparameter $\theta_n$. In practice, we will set $\theta_n = \theta_o$ to select the number of inducing points; the hyperparameter $\theta_n$ is inferred by optimising $\widehat{\mathcal{L}}(M_{\mathbf{b}})$ once the inducing point locations $\mathbf{Z}_n$ have been chosen. In Algorithm 1, $\widehat{\mathcal{U}}(M)$ is used to calculate the stopping criterion. However, in practice, since for the continual learning settings we investigate $N_n$ is small enough, we will use $\widehat{\mathcal{U}}(M_{\mathbf{a}} + N_n) = \mathcal{L}^*$. This value is calculated once at the beginning of the process. The algorithm's complexity depends on the number of inducing points $M_{\mathbf{b}}$ used to compute the lower bound $\widehat{\mathcal{L}}(M_{\mathbf{b}})$ at each iteration. The computational complexity for calculating $\widehat{\mathcal{L}}$ at each batch is $O(N_n M_{\mathbf{b}}^2 + M_{\mathbf{b}}^3)$, and the memory requirement is $O(M_{\mathbf{b}}^2)$, where $M_{\mathbf{b}}$ represents the total number of inducing points in the current batch.

---

**Algorithm 1** Vegas Inducing Point Selection (VIPS)

---

**Input:** $\mathbf{X}_n = \{\mathbf{x}_i\}_{i=1}^{N_n}$, $\mathbf{Z}_o = \{\mathbf{z}_m\}_{m=1}^{M_{\mathbf{a}}}$, $\hat{\mu}, \hat{\sigma}, \theta_n$, kernel $k(\cdot, \cdot | \theta_n)$, threshold parameter $\delta$.
**Output:** Updated set $\mathbf{Z}_n = \mathbf{Z}_o \cup \{\mathbf{x}_{m'}\}_{m'=1}^{M'}$, where $|\mathbf{Z}_n| = M_{\mathbf{b}}$.
Initialise $\mathbf{Z}_n = \mathbf{Z}_o$.
**while** $\widehat{\mathcal{U}}(M) - \widehat{\mathcal{L}}(M_{\mathbf{b}}) \leq \delta |\widehat{\mathcal{U}}(M) - \mathcal{L}_{noise}(\hat{\mu}, \hat{\sigma})|$ **do**
  Select $\mathbf{x} = \operatorname{argmax}_{\mathbf{x} \in \mathbf{X}_n} k(\mathbf{x}, \mathbf{x}) - \mathbf{k_b}(\mathbf{x})^\top \mathbf{K_{bb}^{-1}} \mathbf{k_b}(\mathbf{x})$.
  Add $\mathbf{x}$ to the set of inducing points: $\mathbf{Z}_n = \mathbf{Z}_n \cup \{\mathbf{x}\}$.
**end while**

---

# E    Adaptive Inducing Points Selection Methods

In the experiments, we compare our method, VIPS, to two other adaptive approaches: Conditional Variance (CV) and OIPS [14]. This section contains details about both methods and their implementation.

## E.1    Conditional Variance

The implementation of the Conditional Variance method is presented in Algorithm 2. This method uses the "greedy variance" strategy that iteratively chooses the location of the next inducing point. As a stopping criterion, it uses the trace quantity $\operatorname{tr}(\mathbf{K_{ff}} - \mathbf{Q_{ff}})$. In this algorithm, new inducing points are no longer added once $\operatorname{tr}(\mathbf{K_{ff}} - \mathbf{Q_{ff}})$ falls below a chosen tolerance value $\eta$. Although, this approach was mentioned in Burt et al. [3], this stopping criterion has not yet been tested in the literature. The hyperparameter $\eta$ is determined by the user.

---

**Algorithm 2** Conditional Variance (CV)

---

**Input:** $\mathbf{X}_n = \{\mathbf{x}_i\}_{i=1}^{N_n}$, $\mathbf{Z}_o = \{\mathbf{z}_m\}_{m=1}^{M_{\mathbf{a}}}$, $\theta_n$, kernel, $k(\cdot, \cdot | \theta_n)$, threshold $\eta$.
**Output:** Updated set of inducing points $\mathbf{Z}_n = \{\mathbf{z}_m\}_{m=1}^{M'} \cup \{\mathbf{x}_{m'}\}_{m'=1}^{M_b - M'}$.
Initialise location selection pool: $\mathbf{X}_{pool} = \mathbf{Z}_o \cup \mathbf{X}_m$.
Initialise $\mathbf{Z}_n = \operatorname{argmax}_{\mathbf{x} \in \mathbf{X}_{pool}} k(\mathbf{x}, \mathbf{x})$.
**while** $\operatorname{tr}(\mathbf{K_{ff}} - \mathbf{Q_{ff}}) \leq \eta$ **do**
  Select $\mathbf{x} = \operatorname{argmax}_{\mathbf{x} \in \mathbf{X}_{pool}} k(\mathbf{x}, \mathbf{x}) - \mathbf{k_b}(\mathbf{x})^\top \mathbf{K_{bb}^{-1}} \mathbf{k_b}(\mathbf{x})$.
  Add $\mathbf{x}$ to the set of inducing points: $\mathbf{Z}_n = \mathbf{Z}_n \cup \{\mathbf{x}\}$.
**end while**
**return** $\mathbf{Z}_n$

---

## E.2    Online Inducing Point Selection (OIPS)

Galy-Fajou and Opper [14] introduced the Online Inducing Points Selection (OIPS) algorithm, which iteratively adds points from $\mathbf{X}_n$ to the set of inducing points. The algorithm assesses the impact of each new point on the existing inducing set, based on a covariance threshold. A point $\mathbf{x}$ is added if the maximum value of $\mathbf{k_u}(\mathbf{x})$ falls below a user-defined threshold $\rho$. Algorithm 3 presents our implementation of this method, adapted from the original algorithm in Galy-Fajou and Opper [14].

Table 2 shows a summary of the properties of these methods, as well as the fixed size approach used in Bui et al. [2].

**Algorithm 3** Online Inducing Point Selection (OIPS)

---

**Input:** $\mathbf{X}_n = \{\mathbf{x}_i\}_{i=1}^{N_n}$, $\mathbf{Z}_o = \{\mathbf{z}_m\}_{m=1}^{M_\mathbf{a}}$, kernel function $k(\cdot, \cdot | \theta_n)$, kernel hyperparameters $\theta_n$ (including variance $\sigma_f^2$), acceptance threshold $0 < \rho < 1$.

**Output:** Updated set $\mathbf{Z}_n = \mathbf{Z}_o \cup \{\mathbf{x}_{m'}\}_{m'=1}^{M'}$, where $|\mathbf{Z}_n| = M_\mathbf{b}$.

Initialise $\mathbf{Z}_n = \mathbf{Z}_o$.
Initialise $\bar\rho = \rho \cdot \sigma_f^2$.
**for all** $\mathbf{x}_i \in \mathbf{X}_n$ **do**
    $d = \max_j \left( k\left(\mathbf{x}_i, \mathbf{z}_j | \theta_n \right) \right), \, \forall \mathbf{z}_j \in \mathbf{Z}_n$.
    **if** $d < \bar\rho$ **then**
        Add $\mathbf{x}_i$ to the set of inducing points: $\mathbf{Z}_n = \mathbf{Z}_n \cup \{\mathbf{x}_i\}$.
    **end if**
**end for**

---

Table 2: Properties of inducing points selection method for updating an online GP regression model.

| Method | Type | Selection Pool | Selection Criterion | Stopping Criterion |
|---|---|---|---|---|
| Bui et al. [2] | Fixed | Random sample | Gradient optimisation | M constant |
| Cond. Variance (CV) | Adaptive | $\{X_{new}, Z_{old}\}$ | "greedy variance" | $\mathrm{tr}\left(\mathbf{K_{ff}} - \mathbf{Q_{ff}}\right) \leq \eta$ |
| OIPS [14] | Adaptive | $\{X_{new}\}$ | $\max \mathbf{k_u}(\mathbf{x}) \leq \rho$ | Selection criterion met |
| VIPS | Adaptive | $\{X_{new}\}$ | "greedy variance" | $\widehat{\mathcal{U}} - \widehat{\mathcal{L}} \leq \alpha$ |

# F Further Experimental Details and Results

For all experiments and methods, we use the L-BFGS optimiser.

## F.1 Model size and data distribution

For the synthetic dataset, we generate random noisy observations from the test function $f(x) = \sin(2x) + \cos(5x)$. We used a Squared Exponential kernel initialised with lengthscale $0.5$ and variance $1$. The noise variance was initialised to $0.5$. For VIPS, we use $\delta = 0.05$.

**Dataset 1:** We use $N = 500$ observations uniformly distributed from 0 to 10. The data is ordered and divided into ten batches.

**Dataset 2:** We simulate a scenario where small batches of data are received but the data is distributed across the input space. We use $N = 150$ observations uniformly distributed from 0 to 10. The data is shuffled and divided into ten batches.

**Dataset 3:** We simulate a scenario where only outliers are encountered from time to time and the rest of the data is concentrated around a small part of the input space. We use two sets of data: the first set is sampled from a uniform distribution from 4 to 6, with $N = 1000$ and the second set is sampled from a Cauchy distribution with a mean of $\mu = 5$, with $N = 300$. The data is divided into ten batches, where the first batches only contain observations from the 4 to 6 range and the Cauchy observations are observed in the latter batches.

## F.2 The impact of model capacity in accuracy and training time

### F.2.1 Accuracy comparison

For the synthetic dataset, we generate 1000 random noisy observations from the test function $f(x) = \sin(2x) + \cos(5x)$. We used a Squared Exponential kernel initialised with lengthscale $0.5$ and variance $1.0$. The noise variance was initialised to $0.5$. The performance was measured on a test grid of 500 points. For VIPS, we use $\delta = 0.05$.

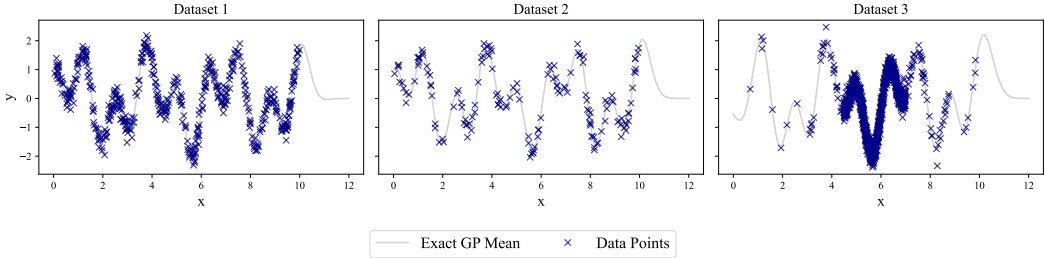

Figure 2: Plot of the three datasets considered in Section F.1.

### F.2.2 Training cost comparison

This experiment was performed on an Nvidia RTX 6000's GPU on a high-performance computing cluster. We used a Squared Exponential kernel with hyperparameters initialised to 1. The noise variance was initialised to 0.1. The dataset was divided into 20 batches, and we recorded the time in training per batch. For VIPS, we use $\delta = 0.05$.

### F.3 UCI datasets

These experiments were performed on an Nvidia RTX 6000's GPU on a high-performance computing cluster. We used a Squared Exponential kernel with hyperparameters initialised to 1 for all datasets. The noise variance was initialised to 0.1. We consider six UCI [8] datasets of different characteristics: Concrete (1030, 8), Skillcraft (3338, 19), Kin8nm (8192, 8), Naval (11934, 14), Elevators (16599, 18), and Bike (17379, 17). The data was sorted by the first dimension. The smaller datasets ($< 12000$) were divided into 20 batches, and the larger ($> 12000$) into 50 batche.

Increasing the model size enhances performance until all relevant dataset information is captured; beyond this point, only computational costs increase. Therefore, when comparing inducing point methods, it is essential to consider the size-performance trade-off, rather than focusing solely on performance gains. This trade-off is typically controlled by a hyperparameter of the model. However, in continual learning, traditional cross-validation for hyperparameter tuning is not feasible, as it would require storing all historical data. Consequently, an effective method must work with hyperparameters that can be set before training and still perform well across diverse datasets. We compare our method, VIPS, with two other adaptive approaches: Conditional Variance (CV) and OIPS. Our goal is to determine if there is a single hyperparameter for each method that performs consistently well across different datasets. To do this, we evaluate various hyperparameter settings for each method and report the Pareto front, showing the trade-off between model size and performance. For each method, we identify the hyperparameter values that achieve a root mean square error (RMSE) within 10% of the full-batch GP across all datasets, which we considered similar to achieving near-exact performance. From these, we select the one that results in the smallest model size. For CV and VIPS, this corresponds to the largest hyperparameter value, while for OIPS, it corresponds to the smallest hyperparameter value (see Table 2 for a summary of the methods). With the optimal hyperparameter selected, all methods achieve the desired performance threshold. Therefore, the preferred method will be the one that minimises model size among the three for each dataset.

Figures 3, 4, 5, 6, 7, 8 present the Pareto fronts for the datasets considered. The selected hyperparameter for each method is highlighted in the plots, indicating the point where each method meets the accuracy constraint while aiming to minimise model size. For CV, $\eta = 0.01$, for OIPS, $\rho = 0.96$, for VIPS $\delta = 0.05$. Table 1 in the main paper shows the specific number of inducing points selected for each method with these particular values at the end of continual learning. VIPS appears to be the preferred method, as its hyperparameter chooses the least inducing points across most datasets. In contrast, CV typically selects the highest number of inducing points. OIPS exhibits variable behaviour, sometimes aligning more closely with CV and other times with VIPS. The hyperparameters for CV and OIPS sometimes lead to excessive model size, for example, in the Kin8nm dataset both methods end up selecting around 80% of the available points, more than double that VIPS.

### F.3.1 Detailed performance for optimal hyperparameters

For the operating hyperparameters, we plot in Figure 9 the selected number of inducing points, RMSE and NLPD versus the number of data points observed throughout the task. The test set for each batch consists only of data from the current and previous batches, therefore we expect the performance metrics to be similar across all batches, provided the model does not experience catastrophic forgetting. As a benchmark, we also plot the exact GP at the first, middle, and last batch of the task, which has access to all observations up to that current batch.

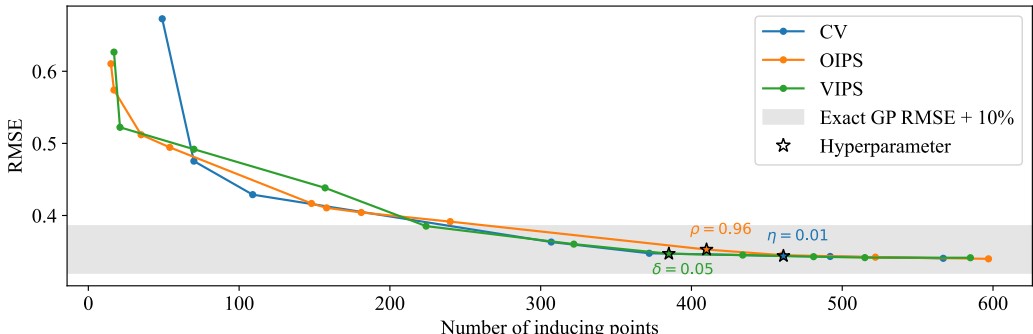

Figure 3: "Concrete" UCI dataset Pareto curve of size and performance for different hyperparameters. for Conditional Variance (range: [0.001, 0.5]), OIPS (range: [0.3, 0.99]) and VIPS (ours, range: [0.001, 4.0]). The plot shows the mean 5-fold test RMSE on the last batch of the task. The shaded region represents the mean 5-fold test RMSE within 10% of the full-batch method. The stars highlight the operating point for each method, with their values listed above.

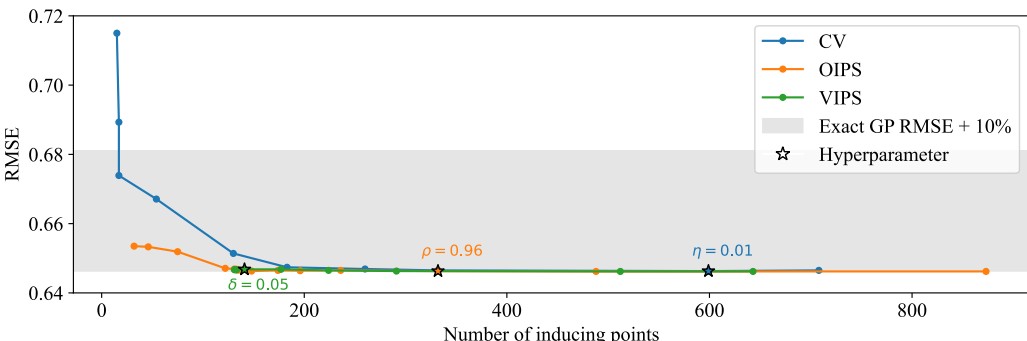

Figure 4: "Skillcraft" UCI dataset Pareto curve of size and performance for different hyperparameters. for Conditional Variance (range: [0.005, 2.0]), OIPS (range: [0.0001, 0.98]) and VIPS (our method, range [0.0005, 1.0], we note that for values of $\delta \geq 1.0$, M stays consistent). The plot shows the mean 5-fold test RMSE on the last batch of the task. The shaded region represents the mean 5-fold test RMSE within 10% of the full-batch method. The stars highlight the operating point for each method, with their values listed above.

### F.3.2 The Naval dataset

For the Naval dataset, Conditional Variance (CV) achieves the accuracy constraint with the fewest inducing points but shows a decline in NLPD throughout the task (Figure 9). OIPS struggles within the tested hyperparameter range [0.96, 0.999], likely due to the dataset's noiseless nature. However, when the data is randomly shuffled (i.i.d. samples), OIPS meets performance requirements with the same hyperparameters (Figure 10), highlighting its sensitivity to data distribution. Despite this, OIPS ends with the highest NLPD, possibly from insufficient inducing points (Table 3). In the i.i.d. setting, CV adds inducing points and improves uncertainty estimates, matching VIPS, which shows minimal sensitivity to distribution changes and robust performance across scenarios. These results emphasise the need for hyperparameters that work consistently across diverse data settings.

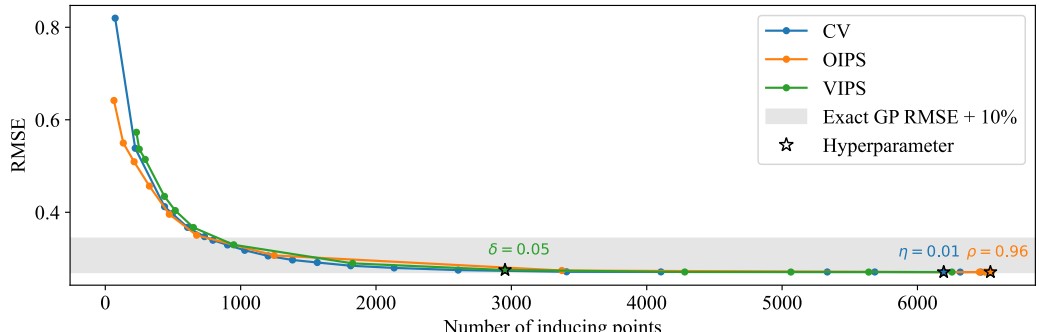

Figure 5: "Kin8nm" UCI dataset Pareto curve of size and performance for different hyperparameters. for Conditional Variance (range: [0.001, 16.0]), OIPS (range: [0.1, 0.97]) and VIPS (ours, range: [0.001, 2.5]). The plot shows the mean 5-fold test RMSE on the last batch of the task. The shaded region represents the mean 5-fold test RMSE within 10% of the full-batch method. The stars highlight the operating point for each method, with their values listed above.

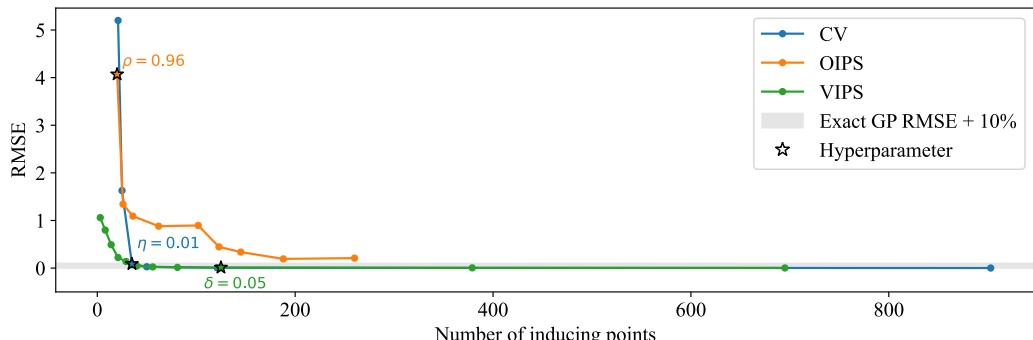

Figure 6: "Naval" UCI dataset Pareto curve of size and performance for different hyperparameters. for Conditional Variance (range: [0.0005, 0.5]), OIPS (range: [0.96, 0.999]) and VIPS (ours, range: [0.01, 5.0]). The plot shows the mean 5-fold test RMSE on the last batch of the task. The shaded region represents the mean 5-fold test RMSE within 10% of the full-batch method. The stars highlight the operating point for each method, with their values listed above.

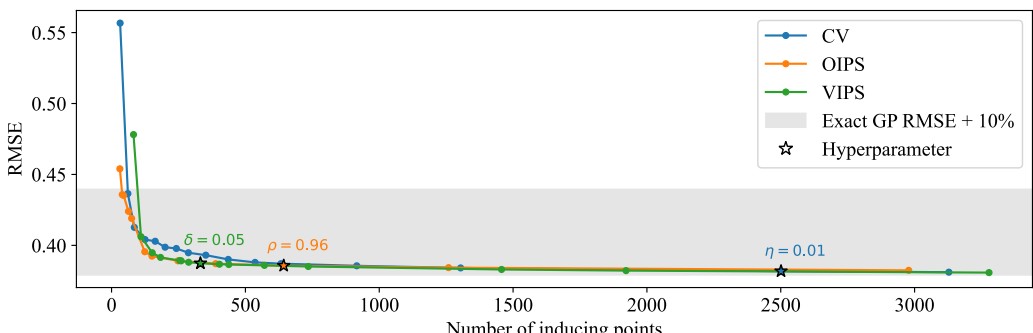

Figure 7: "Elevators" UCI dataset Pareto curve of size and performance for different hyperparameters. for Conditional Variance (range: [0.01, 2.1]), OIPS (range: [0.0001, 0.98]) and VIPS (ours, range: [0.0001, 100]). The plot shows the mean 5-fold test RMSE on the last batch of the task. The shaded region represents the mean 5-fold test RMSE within 10% of the full-batch method. The stars highlight the operating point for each method, with their values listed above.

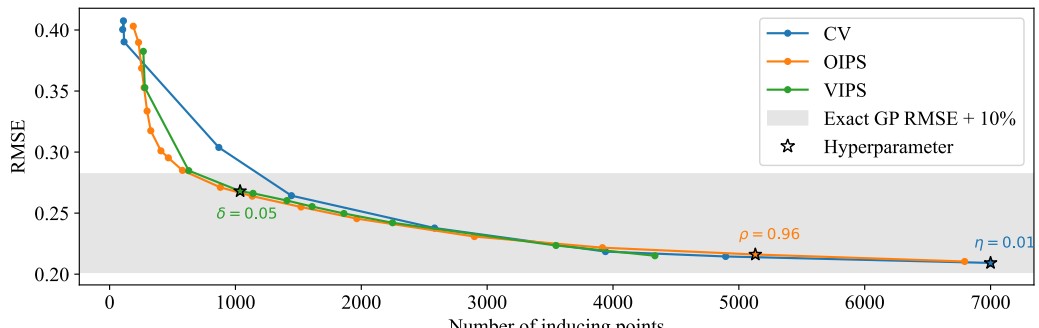

Figure 8: "Bike" UCI dataset Pareto curve of size and performance for different hyperparameters. for Conditional Variance (range: [0.01, 3.0]), OIPS (range: [0.1, 0.97]) and VIPS (ours, range: [0.007, 1.0]). The plot shows the mean 5-fold test RMSE on the last batch of the task. The shaded region represents the mean 5-fold test RMSE within 10% of the full-batch method. The stars highlight the operating point for each method, with their values listed above.

Table 3: Mean (std) of the RMSE, NLPD, and number of inducing points (M) at the end of continual learning on the Naval dataset under two scenarios: data ordered by the first dimension and divided into batches, and data randomly distributed across batches (i.i.d. batches). The table compares the performance of the CV, OIPS, and VIPS methods in both settings, with the Exact GP as benchmark.

| Metrics | Exact GP | CV | OIPS | VIPS | CV i.d.d. | OIPS i.d.d. | VIPS i.d.d. |
|---------|----------|-----|------|------|-----------|-------------|-------------|
| RMSE | 00(.00) | .09(.03) | 5.98(1.28) | .01(.01) | .01(.01) | .04(.01) | .01(.01) |
| NLPD | -5.64(.01) | .98(1.5) | 3.56(.4) | -4.18(.14) | -3.27(.11) | -1.44(.16) | -3.68(.21) |
| M | N/A | 35(3) | 21(1) | 127(5) | 61(2) | 33(1) | 66(6) |

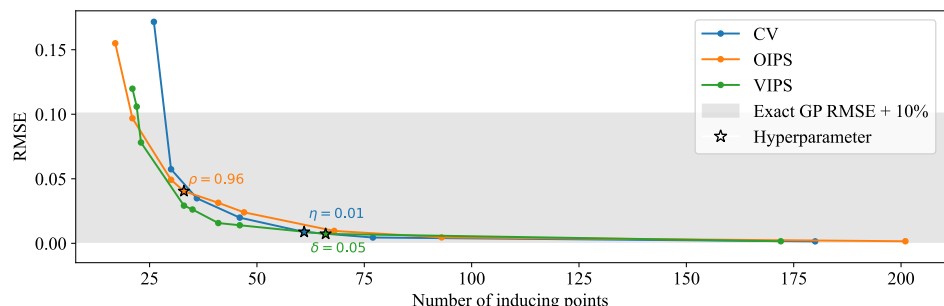

Figure 10: "Naval" UCI dataset Pareto curve of size and performance for different hyperparameters when data is randomly distributed across batches (i.i.d. batches), for Conditional Variance (range: [0.001, 0.2]), OIPS (range: [0.8, 0.999]) and VIPS (ours, range: [0.03, 4.0]). The plot shows the mean 5-fold test RMSE on the last batch of the task. The shaded region represents the mean 5-fold test RMSE within 10% of the full-batch method. The stars highlight the operating point for each method, with their values listed above.

## F.4 Magnetic anomalies

The data used in this experiment is obtained from Solin et al. [39] and is available on GitHub. The objective of this task is to detect local anomalies in the Earth's magnetic field online, caused by the presence of bedrock and magnetic materials in indoor building structures. For this purpose, a small robot with a 3-axis magnetometer moves around an indoor space of approximately 6 meters by 6 meters and measures the magnetic field strength. Out of the 9 available trajectories, we use trajectories 1, 2, 3, 4, and 5 (with $n = 8875, 9105, 9404, 7332, 8313$, respectively) for the

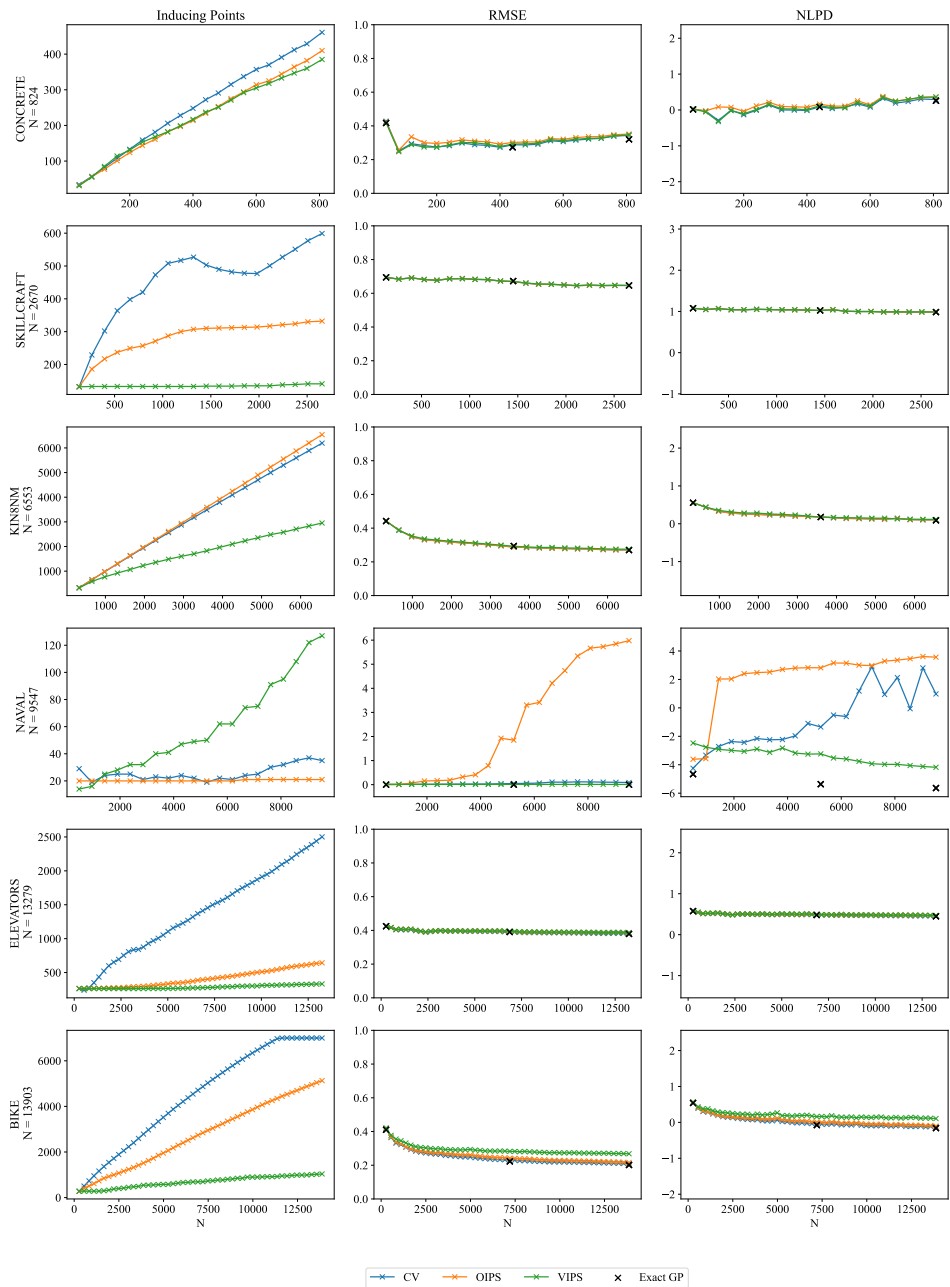

Figure 9: Plot of the mean number of inducing points, RMSE and NLPD over five 80%-train 20%-test random splits versus the number of data points, denoted by N, observed up to and including the current batch for different inducing point selection methods. The corresponding dataset is indicated at the left of the graph and each column corresponds to one of the metrics. The exact GP has access to all data up to and including the current batch.

experiments. Specifically, we use trajectories 1, 2, 4 and 5 for the first experiment and trajectory 3 for the second.

We use the experimental setup proposed in Chang et al. [4]. The proposed model applies a GP prior to magnetic field strength, given by $\mathcal{GP}\left(0, \sigma_0^2 + \kappa_{\sigma^2,\ell}^{\mathrm{Mat}}(\mathbf{x}, \mathbf{x}')\right)$ (in $\mu$T ), where the kernel consist of a constant kernel and a Matérn-$\nu/2$ kernel. The model assumes the spatial domain is affected by Gaussian noise with a variance $\sigma_n^2$. The initial variance for the constant kernel is set to $500$, and the Gaussian likelihood is initialised with a noise variance of $0.1$.

Our aim for this experiment is to test the optimal hyperparameters identified in the previous section for each adaptive method in a real-world setting. The setting simulates an ever-expanding domain, where the robot is not confined to a predefined area. In this context, the model continuously learns new parts of the space. Therefore, a method that works will need to sufficiently expand the model's size to accommodate new data without letting it grow uncontrollably.

In the first experiment, we aim to sequentially learn the paths taken by the robot using trajectories 1, 2, 4 and 5, i.e. an entire path will correspond to one batch. We investigate whether the method can adapt to changes in the environment and adjust the number of inducing points accordingly. During this process, we concurrently learn the hyperparameters $\sigma_0^2$, $\sigma^2$, $\ell$, and $\sigma_n^2$. As a test set, we use trajectory 3. Figures 11, 13, 15 show the temporally updating field estimate over batches alongside the corresponding path travelled in each batch.

In the second experiment, we focus on the streaming learning of trajectory 3. The trajectory is split into 20 batches. We compare the number of inducing points selected and the estimate obtained by the three methods, Conditional Variance (CV), OIPS and VIPS (ours). Detailed learning of the path for each method is shown in figures 12, 14 and 16. As a test set, we use trajectories 1, 2, 4, and 5. We observed how Conditional Variance chooses an excessive number of inducing points, indicating that its hyperparameter needs tuning, which is impractical in the continual learning setting. OIPS selects the fewest inducing points, concentrating them at the start of the path and becoming sparse towards the end. However, its estimates differ significantly from those in the previous experiment, indicating that it fails to add sufficient capacity to capture changes in the environment. VIPS provides the middle ground, selecting a moderate number of inducing points that effectively balance accuracy and memory size. This choice allows VIPS to maintain a robust estimate of the magnetic field obtained when compared to learning by paths without excessive computational overhead.

In the last two sections, when compared to both alternative approaches, VIPS achieved the best trade-off between performance and model size without requiring hyperparameter tuning, making it the preferred method among the three.

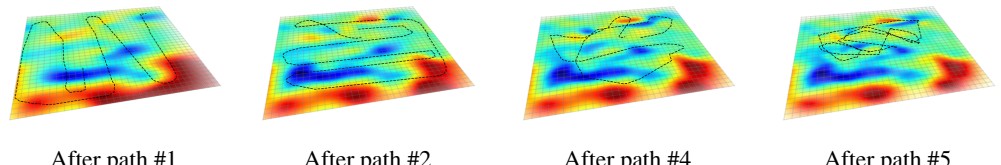

| After path #1 | After path #2 | After path #4 | After path #5 |

Figure 11: VIPS (Ours). A small robot with wheels is used to perform sequential estimation of magnetic field anomalies. We show the estimate of the magnitude field learned sequentially after travelling the path shown in a dotted line. The degree of transparency represents the marginal variance.

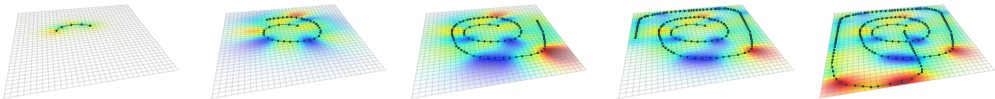

Batch 1, M = 5.     Batch 5, M = 35.     Batch 10, M = 80.     Batch 15, M = 133.  Batch 20, M = 174.

Figure 12: VIPS (Ours). A small robot with wheels is used to perform sequential estimation of magnetic field anomalies. Data is collected continuously as the robot moves along the path. The inducing points are represented as black dots and the line represents the travelled part of the path. We indicate the batch number and number of inducing points (M). Final RMSE = 7.59.

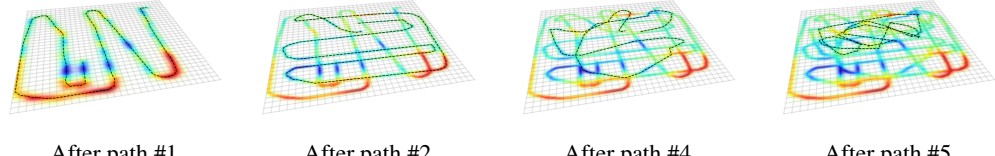

After path #1      After path #2      After path #4      After path #5

Figure 13: Conditional Variance. A small robot with wheels is used to perform sequential estimation of magnetic field anomalies. We show the estimate of the magnitude field learned sequentially after travelling the path shown in a dotted line. The degree of transparency represents the marginal variance.

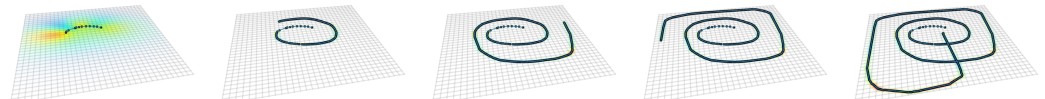

Batch 1, M = 12.     Batch 5, M = 597.   Batch 10, M = 2206.  Batch 15, M = 4071.  Batch 20, M = 5000.

Figure 14: Conditional Variance. A small robot with wheels is used to perform sequential estimation of magnetic field anomalies. Data is collected continuously as the robot moves along the path. The inducing points are represented as black dots and the line represents the travelled part of the path. We indicate the batch number and number of inducing points (M). Final RMSE = 10.66.

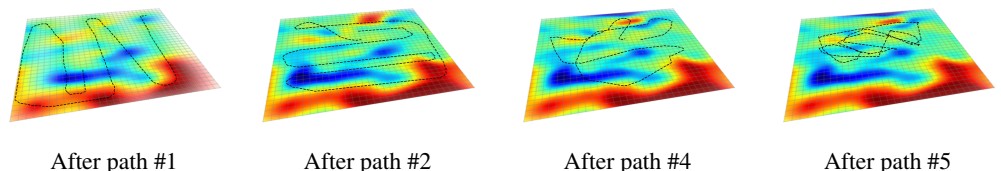

After path #1      After path #2      After path #4      After path #5

Figure 15: OIPS. A small robot with wheels is used to perform sequential estimation of magnetic field anomalies. We show the estimate of the magnitude field learned sequentially after travelling the path shown in a dotted line. The degree of transparency represents the marginal variance.

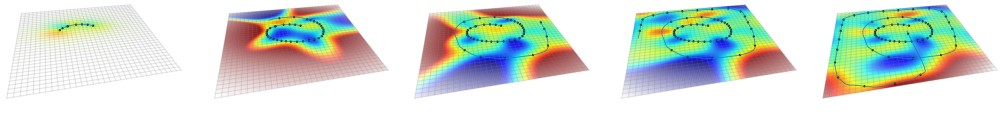

Batch 1, M = 15.     Batch 5, , M = 43.   Batch 10, M = 52.     Batch 15, M = 63.     Batch 20, M = 71.

Figure 16: OIPS. A small robot with wheels is used to perform sequential estimation of magnetic field anomalies. Data is collected continuously as the robot moves along the path. The inducing points are represented as black dots and the line represents the travelled part of the path. We indicate the batch number and number of inducing points (M). Final RMSE = 9.81.

# G Further derivations and implementation details

## G.1 Derivation of $\mathcal{L}^*$

Recall that

$$\mathcal{L}^* = \log \mathcal{N}\left(\hat{\mathbf{y}};\, \mathbf{0}\,, \mathbf{K}_{\hat{\mathbf{f}}\hat{\mathbf{f}}} + \Sigma_{\hat{y}}\right) + \Delta_{\mathbf{a}}, \tag{23}$$

with $\mathbf{K}_{\hat{\mathbf{f}}\hat{\mathbf{f}}} = \begin{bmatrix} \mathbf{K}_{\mathbf{ff}} & \mathbf{K}_{\mathbf{fa}} \\ \mathbf{K}_{\mathbf{af}} & \mathbf{K}_{\mathbf{aa}} \end{bmatrix}$.

The first term can be lower bounded using Jensen's inequality as,

$$\log \mathcal{N}\left(\hat{\mathbf{y}};\, \mathbf{0}\,, \mathbf{K}_{\hat{\mathbf{f}}\hat{\mathbf{f}}} + \Sigma_{\hat{y}}\right)$$
$$\geq \log \mathcal{N}\left(\hat{\mathbf{y}};\, \mathbf{0}\,, \mathbf{K}_{\hat{\mathbf{f}}\mathbf{b}}\mathbf{K}_{\mathbf{bb}}^{-1}\mathbf{K}_{\mathbf{b}\hat{\mathbf{f}}} + \Sigma_{\hat{y}}\right) - \frac{1}{2}\operatorname{tr}\left(\Sigma_{\hat{\mathbf{y}}}^{-1}\left(\mathbf{K}_{\hat{\mathbf{f}}\hat{\mathbf{f}}} - \mathbf{K}_{\hat{\mathbf{f}}\mathbf{b}}\mathbf{K}_{\mathbf{bb}}^{-1}\mathbf{K}_{\mathbf{b}\hat{\mathbf{f}}}\right)\right) \tag{24}$$

where $\mathbf{K}_{\hat{\mathbf{f}}\mathbf{b}}\mathbf{K}_{\mathbf{bb}}^{-1}\mathbf{K}_{\mathbf{b}\hat{\mathbf{f}}}$ is the Nyström approximation of $\mathbf{K}_{\hat{\mathbf{f}}\hat{\mathbf{f}}}$. The trace will be small when $\mathbf{b} = \{f(\mathbf{x}_{new}, \mathbf{a})\}$ and can be simplified as follows:

$$\operatorname{tr}\left(\Sigma_{\hat{\mathbf{y}}}^{-1}\left(\mathbf{K}_{\hat{\mathbf{f}}\hat{\mathbf{f}}} - \mathbf{K}_{\hat{\mathbf{f}}\mathbf{b}}\mathbf{K}_{\mathbf{bb}}^{-1}\mathbf{K}_{\mathbf{b}\hat{\mathbf{f}}}\right)\right)$$
$$= \operatorname{tr}\left(\begin{bmatrix} \sigma_y^{-2}\mathbf{I} & \mathbf{0} \\ \mathbf{0} & \mathbf{D}_{\mathbf{a}}^{-1} \end{bmatrix}\left(\begin{bmatrix} \mathbf{K}_{\mathbf{ff}} & \mathbf{K}_{\mathbf{fa}} \\ \mathbf{K}_{\mathbf{af}} & \mathbf{K}_{\mathbf{aa}} \end{bmatrix} - \begin{bmatrix} \mathbf{K}_{\mathbf{fb}}\mathbf{K}_{\mathbf{bb}}^{-1}\mathbf{K}_{\mathbf{bf}} & \mathbf{K}_{\mathbf{fb}}\mathbf{K}_{\mathbf{bb}}^{-1}\mathbf{K}_{\mathbf{ba}} \\ \mathbf{K}_{\mathbf{ab}}\mathbf{K}_{\mathbf{bb}}^{-1}\mathbf{K}_{\mathbf{bf}} & \mathbf{K}_{\mathbf{ab}}\mathbf{K}_{\mathbf{bb}}^{-1}\mathbf{K}_{\mathbf{ba}} \end{bmatrix}\right)\right) \tag{25}$$
$$= \operatorname{tr}\left(\mathbf{D}_{\mathbf{a}}^{-1}(\mathbf{K}_{\mathbf{aa}} - \mathbf{K}_{\mathbf{ab}}\mathbf{K}_{\mathbf{bb}}^{-1}\mathbf{K}_{\mathbf{ba}})\right) + \frac{1}{\sigma^2}\operatorname{tr}(\mathbf{K}_{\mathbf{ff}} - \mathbf{K}_{\mathbf{fb}}\mathbf{K}_{\mathbf{bb}}^{-1}\mathbf{K}_{\mathbf{bf}})$$

which recovers the expression for $\widehat{\mathcal{L}} - \Delta_{\mathbf{a}}$.

## G.2 Online Upper Bound Implementation

In this section, we provide efficient forms for practical implementation of the online upper bound $\widehat{\mathcal{U}}$. As the second term is constant we focus on the first term,

$$\widehat{\mathcal{U}}_2 = -\frac{(N + M_{\mathbf{a}})}{2}\log(2\pi) - \frac{1}{2}\log|\mathbf{K}_{\hat{\mathbf{f}}\mathbf{b}}\mathbf{K}_{\mathbf{bb}}^{-1}\mathbf{K}_{\mathbf{b}\hat{\mathbf{f}}} + \Sigma_{\hat{\mathbf{y}}}| - \frac{1}{2}\hat{\mathbf{y}}^T\left(\mathbf{K}_{\hat{\mathbf{f}}\mathbf{b}}\mathbf{K}_{\mathbf{bb}}^{-1}\mathbf{K}_{\mathbf{b}\hat{\mathbf{f}}} + t\mathbf{I} + \Sigma_{\hat{\mathbf{y}}}\right)^{-1}\hat{\mathbf{y}}. \tag{26}$$

This term is an upper bound for the first term of $\mathcal{L}^* = \log \mathcal{N}\left(\hat{\mathbf{y}};\, \mathbf{0}\,, \mathbf{K}_{\hat{\mathbf{f}}\hat{\mathbf{f}}} + \Sigma_{\hat{y}}\right) + \Delta_{\mathbf{a}}$.

### G.2.1 Determinant term

Letting $\mathbf{K}_{\mathbf{bb}} = \mathbf{L}_{\mathbf{b}}\mathbf{L}_{\mathbf{b}}^T$ and using the matrix determinant lemma, we can rewrite the determinant term as

$$\log|\mathbf{K}_{\hat{\mathbf{f}}\mathbf{b}}\mathbf{K}_{\mathbf{bb}}^{-1}\mathbf{K}_{\mathbf{b}\hat{\mathbf{f}}} + \Sigma_{\hat{\mathbf{y}}}| = \log|\Sigma_{\hat{\mathbf{y}}}| + \log|\mathbf{I} + \mathbf{L}_{\mathbf{b}}^{-1}\mathbf{K}_{\mathbf{b}\hat{\mathbf{f}}}\Sigma_{\hat{\mathbf{y}}}^{-1}\mathbf{K}_{\hat{\mathbf{f}}\mathbf{b}}\mathbf{L}_{\mathbf{b}}^{-T}|$$
$$= N\log\sigma_y^2 + \log|\mathbf{D}_{\mathbf{a}}| + \log|\mathbf{I} + \mathbf{L}_{\mathbf{b}}^{-1}\mathbf{K}_{\mathbf{b}\hat{\mathbf{f}}}\Sigma_{\hat{\mathbf{y}}}^{-1}\mathbf{K}_{\hat{\mathbf{f}}\mathbf{b}}\mathbf{L}_{\mathbf{b}}^{-T}| \tag{27}$$

Let $\mathbf{D} = \mathbf{I} + \mathbf{L}_{\mathbf{b}}^{-1}\mathbf{K}_{\mathbf{b}\hat{\mathbf{f}}}\Sigma_{\hat{\mathbf{y}}}^{-1}\mathbf{K}_{\hat{\mathbf{f}}\mathbf{b}}\mathbf{L}_{\mathbf{b}}^{-T}$. Note that,

$$\mathbf{K}_{\mathbf{b}\hat{\mathbf{f}}}\Sigma_{\hat{\mathbf{y}}}^{-1}\mathbf{K}_{\hat{\mathbf{f}}\mathbf{b}} = \begin{bmatrix} \mathbf{K}_{\mathbf{bf}} & \mathbf{K}_{\mathbf{ba}} \end{bmatrix}\begin{bmatrix} \frac{1}{\sigma_y^2}\mathbf{I} & 0 \\ 0 & \mathbf{D}_{\mathbf{a}}^{-1} \end{bmatrix}\begin{bmatrix} \mathbf{K}_{\mathbf{fb}} \\ \mathbf{K}_{\mathbf{ab}} \end{bmatrix}$$
$$= \frac{1}{\sigma_y^2}\mathbf{K}_{\mathbf{bf}}\mathbf{K}_{\mathbf{fb}} + \mathbf{K}_{\mathbf{ba}}\mathbf{D}_{\mathbf{a}}^{-1}\mathbf{K}_{\mathbf{ab}} \tag{28}$$
$$= \frac{1}{\sigma_y^2}\mathbf{K}_{\mathbf{bf}}\mathbf{K}_{\mathbf{fb}} + \mathbf{K}_{\mathbf{ba}}\mathbf{S}_{\mathbf{a}}^{-1}\mathbf{K}_{\mathbf{ab}} - \mathbf{K}_{\mathbf{ba}}\mathbf{K}_{\mathbf{aa}}'^{-1}\mathbf{K}_{\mathbf{ab}}.$$

Therefore,

$$\mathbf{D} = \mathbf{I} + \frac{1}{\sigma_y^2}\mathbf{L}_{\mathbf{b}}^{-1}\mathbf{K}_{\mathbf{bf}}\mathbf{K}_{\mathbf{fb}}\mathbf{L}_{\mathbf{b}}^{-T} + \mathbf{L}_{\mathbf{b}}^{-1}\mathbf{K}_{\mathbf{ba}}\mathbf{S}_{\mathbf{a}}^{-1}\mathbf{K}_{\mathbf{ab}}\mathbf{L}_{\mathbf{b}}^{-T} - \mathbf{L}_{\mathbf{b}}^{-1}\mathbf{K}_{\mathbf{ba}}\mathbf{K}_{\mathbf{aa}}'^{-1}\mathbf{K}_{\mathbf{ab}}\mathbf{L}_{\mathbf{b}}^{-T}. \tag{29}$$

### G.2.2 Quadratic term

Given the quadratic term,

$$-\frac{1}{2}\hat{\mathbf{y}}^T\left(\mathbf{K}_{\hat{\mathbf{f}}\mathbf{b}}\mathbf{K}_{\mathbf{bb}}^{-1}\mathbf{K}_{\mathbf{b}\hat{\mathbf{f}}} + t\mathbf{I} + \Sigma_{\hat{\mathbf{y}}}\right)^{-1}\hat{\mathbf{y}}.$$

Letting $\widehat{\Sigma}_{\hat{\mathbf{y}}} = t\mathbf{I} + \Sigma_{\hat{\mathbf{y}}}$ and by Woodbury's formula, we obtain:

$$\left(\mathbf{K}_{\hat{\mathbf{f}}\mathbf{b}}\mathbf{K}_{\mathbf{bb}}^{-1}\mathbf{K}_{\mathbf{b}\hat{\mathbf{f}}} + \widehat{\Sigma}_{\hat{\mathbf{y}}}\right)^{-1} = \widehat{\Sigma}_{\hat{\mathbf{y}}}^{-1} - \widehat{\Sigma}_{\hat{\mathbf{y}}}^{-1}\mathbf{K}_{\hat{\mathbf{f}}\mathbf{b}}\mathbf{L}_{\mathbf{b}}^{-T}\left(I + \mathbf{L}_{\mathbf{b}}^{-1}\mathbf{K}_{\mathbf{b}\hat{\mathbf{f}}}\widehat{\Sigma}_{\hat{\mathbf{y}}}^{-1}\mathbf{K}_{\hat{\mathbf{f}}\mathbf{b}}\mathbf{L}_{\mathbf{b}}^{-T}\right)^{-1}\mathbf{L}_{\mathbf{b}}^{-1}\mathbf{K}_{\mathbf{b}\hat{\mathbf{f}}}\widehat{\Sigma}_{\hat{\mathbf{y}}}^{-1}.$$

We have,

$$\hat{\mathbf{y}}^T\widehat{\Sigma}_{\hat{\mathbf{y}}}^{-1}\hat{\mathbf{y}} = \frac{1}{\sigma_y^2 + t}\mathbf{y}^T\mathbf{y} + \left(\mathbf{D}_{\mathbf{a}}\mathbf{S}_{\mathbf{a}}^{-1}\mathbf{m}_{\mathbf{a}}\right)^T\left(\mathbf{D}_{\mathbf{a}} + tI\right)^{-1}\left(\mathbf{D}_{\mathbf{a}}\mathbf{S}_{\mathbf{a}}^{-1}\mathbf{m}_{\mathbf{a}}\right). \tag{30}$$

Letting $\widehat{\mathbf{D}} = \mathbf{I} + \mathbf{L}_{\mathbf{b}}^{-1}\mathbf{K}_{\mathbf{b}\hat{\mathbf{f}}}\widehat{\Sigma}_{\hat{\mathbf{y}}}^{-1}\mathbf{K}_{\hat{\mathbf{f}}\mathbf{b}}\mathbf{L}_{\mathbf{b}}^{-T}$ where,

$$\mathbf{K}_{\mathbf{b}\hat{\mathbf{f}}}\Sigma_{\mathbf{c}}^{-1}\mathbf{K}_{\hat{\mathbf{f}}\mathbf{b}} = \begin{bmatrix}^\top\mathbf{K}_{\mathbf{fb}}\\\mathbf{K}_{\mathbf{ab}}\end{bmatrix}\begin{bmatrix}\frac{1}{\sigma_y^2+t}\mathbf{I} & 0\\0 & (\mathbf{D}_{\mathbf{a}}+t\mathbf{I})^{-1}\end{bmatrix}\begin{bmatrix}\mathbf{K}_{\mathbf{fb}}\\\mathbf{K}_{\mathbf{ab}}\end{bmatrix} = \frac{1}{\sigma_y^2+t}\mathbf{K}_{\mathbf{bf}}\mathbf{K}_{\mathbf{fb}} + \mathbf{K}_{\mathbf{ba}}(\mathbf{D}_{\mathbf{a}}+t\mathbf{I})^{-1}\mathbf{K}_{\mathbf{ab}}$$

and letting $\hat{\mathbf{c}} = \mathbf{K}_{\mathbf{b}\hat{\mathbf{f}}}\widehat{\Sigma}_{\hat{\mathbf{y}}}^{-1}\hat{\mathbf{y}} = \frac{1}{\sigma_y^2+t}\mathbf{K}_{\mathbf{bf}}\mathbf{y} + \mathbf{K}_{\mathbf{ba}}\left(\mathbf{D}_{\mathbf{a}}+t\mathbf{I}\right)^{-1}\left(\mathbf{D}_{\mathbf{a}}\mathbf{S}_{\mathbf{a}}^{-1}\mathbf{m}_{\mathbf{a}}\right)$, we obtain,

$$\hat{\mathbf{y}}^T\widehat{\Sigma}_{\hat{\mathbf{y}}}^{-1}\mathbf{K}_{\hat{\mathbf{f}}\mathbf{b}}\mathbf{L}_{\mathbf{b}}^{-T}\left(I + \mathbf{L}_{\mathbf{b}}^{-1}\mathbf{K}_{\mathbf{b}\hat{\mathbf{f}}}\widehat{\Sigma}_{\hat{\mathbf{y}}}^{-1}\mathbf{K}_{\hat{\mathbf{f}}\mathbf{b}}\mathbf{L}_{\mathbf{b}}^{-T}\right)^{-1}\mathbf{L}_{\mathbf{b}}^{-1}\mathbf{K}_{\mathbf{b}\hat{\mathbf{f}}}\widehat{\Sigma}_{\hat{\mathbf{y}}}^{-1}\hat{\mathbf{y}} = \hat{\mathbf{c}}^T\mathbf{L}_{\mathbf{b}}^{-T}\widehat{\mathbf{D}}^{-1}\mathbf{L}_{\mathbf{b}}^{-1}\hat{\mathbf{c}}. \tag{31}$$

Putting this back into the upper bound:

$$\widehat{\mathcal{U}}_2 = -\frac{(N+M_{\mathbf{a}})}{2}\log(2\pi) - \frac{1}{2}N\log\sigma_y^2 - \frac{1}{2}\log|\mathbf{D}_{\mathbf{a}}| - \frac{1}{2}\log|\mathbf{D}| - \frac{1}{2}\hat{\mathbf{y}}^T\widehat{\Sigma}_{\hat{\mathbf{y}}}^{-1}\hat{\mathbf{y}} + \frac{1}{2}\hat{\mathbf{c}}^T\mathbf{L}_{\mathbf{b}}^{-T}\widehat{\mathbf{D}}^{-1}\mathbf{L}_{\mathbf{b}}^{-1}\hat{\mathbf{c}}. \tag{32}$$

The upper bound for $\mathcal{L}^*$ is therefore

$$\widehat{\mathcal{U}} = -\frac{(N+M_{\mathbf{a}})}{2}\log(2\pi) - \frac{1}{2}\log|\mathbf{K}_{\hat{\mathbf{f}}\mathbf{b}}\mathbf{K}_{\mathbf{bb}}^{-1}\mathbf{K}_{\mathbf{b}\hat{\mathbf{f}}} + \Sigma_{\hat{\mathbf{y}}}| - \frac{1}{2}\hat{\mathbf{y}}^T\left(\mathbf{K}_{\hat{\mathbf{f}}\mathbf{b}}\mathbf{K}_{\mathbf{bb}}^{-1}\mathbf{K}_{\mathbf{b}\hat{\mathbf{f}}} + t\mathbf{I} + \Sigma_{\hat{\mathbf{y}}}\right)^{-1}\hat{\mathbf{y}} + \Delta_{\mathbf{a}}$$

$$= -\frac{N}{2}\log(2\pi\sigma_y^2) - \frac{1}{2}\log|\mathbf{D}| - \frac{1}{2}\frac{1}{\sigma_y^2+t}\mathbf{y}^T\mathbf{y} - \frac{1}{2}\left(\mathbf{D}_{\mathbf{a}}\mathbf{S}_{\mathbf{a}}^{-1}\mathbf{m}_{\mathbf{a}}\right)^T\left(\mathbf{D}_{\mathbf{a}} + tI\right)^{-1}\left(\mathbf{D}_{\mathbf{a}}\mathbf{S}_{\mathbf{a}}^{-1}\mathbf{m}_{\mathbf{a}}\right)$$

$$+ \frac{1}{2}\hat{\mathbf{c}}^T\mathbf{L}_{\mathbf{b}}^{-T}\widehat{\mathbf{D}}^{-1}\mathbf{L}_{\mathbf{b}}^{-1}\hat{\mathbf{c}} - \frac{1}{2}\log\frac{|\mathbf{S}_{\mathbf{a}}|}{|\mathbf{K}'_{\mathbf{aa}}|} - \frac{1}{2}\mathbf{m}_{\mathbf{a}}^T\mathbf{S}_{\mathbf{a}}^{-1}\mathbf{m}_{\mathbf{a}} + \frac{1}{2}\mathbf{m}_{\mathbf{a}}^T\mathbf{S}_{\mathbf{a}}^{-1}\mathbf{D}_{\mathbf{a}}\mathbf{S}_{\mathbf{a}}^{-1}\mathbf{m}_{\mathbf{a}}. \tag{33}$$

