# OpenReview forum: "Adjusting Model Size in Continual Gaussian Processes: How Big is Big Enough?"
_NeurIPS.cc/2024/Workshop/BDU — NeurIPS BDU Workshop 2024 Poster_

### Official Review · Reviewer_T22N · 2024-09-27
**review on paper43**

**Rating:** 9
**Confidence:** 4

**Review:**

The paper presents an innovative approach to adjusting model size adaptively for Gaussian processes in continual learning settings. This problem is significant due to the dynamic nature of data availability in many real-world applications, and the solution proposed addresses the need for efficiency and scalability without sacrificing performance.

Novelty and Relevance: The concept of adaptive model sizing based on incoming data is novel and very relevant to current trends in machine learning, particularly in areas dealing with streaming data.

Methodological Rigor: The paper is methodologically sound, detailing the criteria for model adjustment and integrating these into the Gaussian process framework effectively.

---

### Official Review · Reviewer_3F3r · 2024-10-03

**Rating:** 6
**Confidence:** 4

**Review:**

This paper proposes an algorithm that automatically adjusts the number of data points (inducing points) to use while maintaining near-optimal performance for Gaussian Processes (single-layer neural networks) in continual learning. The algorithm dynamically selects inducing points based on the data, ensuring efficient computation and strong predictive accuracy. Notably, it requires tuning only a single hyperparameter and performs well across various datasets.

Strengths:
1. The proposed algorithm is highly valuable in the context of streaming data. With only one hyperparameter to tune, it significantly reduces the need for fine-tuning, making the contribution noteworthy.
2. The empirical results are convincing and effectively demonstrate the algorithm's performance and efficacy across various datasets.

Weaknesses:
1. The paper could benefit from comparing the proposed algorithm with other adaptive methods in terms of prediction efficiency and additional metrics such as computational efficiency.
2. There is a lack of explanation regarding the underlying motivation for why the algorithm is effective in the continual learning and streaming data setting, which would enhance the understanding of its strengths.

---

### Decision · Program_Chairs · 2024-10-09

Accept (Poster)